# Generalization Guarantees for Learning Score-Based Branch-and-Cut Policies in Integer Programming

**Hongyu Cheng**
Dept. of Applied Mathematics & Statistics
Johns Hopkins University
Baltimore, MD 21218
hongyucheng@jhu.edu

**Amitabh Basu**
Dept. of Applied Mathematics & Statistics
Johns Hopkins University
Baltimore, MD 21218
basu.amitabh@jhu.edu

## Abstract

Mixed-integer programming (MIP) provides a powerful framework for optimization problems, with Branch-and-Cut (B&C) being the predominant algorithm in state-of-the-art solvers. The efficiency of B&C critically depends on heuristic policies for making sequential decisions, including node selection, cut selection, and branching variable selection. While traditional solvers often employ heuristics with manually tuned parameters, recent approaches increasingly leverage machine learning, especially neural networks, to learn these policies directly from data. A key challenge is to understand the theoretical underpinnings of these learned policies, particularly their generalization performance from finite data. This paper establishes rigorous sample complexity bounds for learning B&C policies where the scoring functions guiding each decision step (node, cut, branch) have a certain piecewise polynomial structure. This structure generalizes the linear models that form the most commonly deployed policies in practice and investigated recently in a foundational series of theoretical works by Balcan et al. Such piecewise polynomial policies also cover the neural network architectures (e.g., using ReLU activations) that have been the focal point of contemporary practical studies. Consequently, our theoretical framework closely reflects the models utilized by practitioners investigating machine learning within B&C, offering a unifying perspective relevant to both established theory and modern empirical research in this area. Furthermore, our theory applies to quite general sequential decision making problems beyond B&C.

## 1 Introduction

Mixed-Integer Programming (MIP) provides a powerful optimization tool for problems arising in diverse fields such as finance [Cornuejols et al., 2018], vehicle routing [Toth and Vigo, 2014], computational and systems biology [Gusfield, 2019], telecommunications network design [Kerivin and Mahjoub, 2005], production planning [Pochet and Wolsey, 2006], to name a few, where decisions involve both discrete choices and continuous adjustments. We focus on problems formulated as $\min\left\{\mathbf{c}^{\mathsf{T}}\mathbf{x} \mid A\mathbf{x} \leq \mathbf{b}, \mathbf{x} \in \mathbb{Z}_+^{n_1} \times \mathbb{R}_+^{n_2}\right\}$, where $A \in \mathbb{Q}^{m \times (n_1+n_2)}$, $\mathbf{b} \in \mathbb{Q}^m$, and $\mathbf{c} \in \mathbb{R}^{n_1+n_2}$. The predominant methodology for solving such MIPs is the Branch-and-Cut (B&C) algorithm [Jünger et al., 2009, Conforti et al., 2014] (see Algorithm 1 below for a standard outline). B&C operates by exploring a search tree where each node represents a linear programming (LP) relaxation of a subproblem derived from the original MIP. The process involves solving these LP relaxations, adding valid inequalities (cutting planes) to tighten the relaxations without excluding feasible integer solutions, and partitioning the solution space (branching) by imposing constraints on variables, typically integer-constrained variables ($j \in [n_1]$) that take fractional values in an LP solution. The practical performance of the B&C algorithm is critically influenced by the sequence of strategic

39th Conference on Neural Information Processing Systems (NeurIPS 2025).

decisions made during this dynamic process, including how nodes are selected for exploration, which cutting planes are added, and which variables are chosen for branching [Achterberg et al., 2005].

---

**Algorithm 1** The Branch-and-Cut Algorithm

---

**Require:** Initial MIP instance $I$, maximum rounds $M$, error tolerance $\epsilon_{\text{gap}}$.
1: Initialize open node list $\mathcal{L} \leftarrow \{\mathsf{N}_0\}$, $\mathsf{UB} \leftarrow \infty$, $\mathsf{LB} \leftarrow -\infty$, $\mathbf{x}^* \leftarrow$ null, $i \leftarrow 0$.
2: **while** $\mathcal{L} \neq \emptyset$ and $\mathsf{UB} - \mathsf{LB} > \epsilon_{\text{gap}}$ and $i < M$ **do**
3:     **Node Selection:** Choose $\mathsf{N}^* \in \mathcal{L}$ via node selection policy; $\mathcal{L} \leftarrow \mathcal{L} \setminus \{\mathsf{N}^*\}$.
4:     Solve LP at $\mathsf{N}^*$; let solution be $\mathbf{x}_{\mathsf{LP}}^{\mathsf{N}^*}$ and value be $z_{\mathsf{LP}}^{\mathsf{N}^*}$.
5:     Let $z_{\mathsf{LP}}^{\mathsf{N}}$ denote the LP value associated with any node $\mathsf{N} \in \mathcal{L}$.
6:     **if** LP infeasible or $z_{\mathsf{LP}}^{\mathsf{N}^*} \geq \mathsf{UB}$ **then goto** Line 18
7:     **end if**
8:     **if** $\mathbf{x}_{\mathsf{LP}}^{\mathsf{N}^*}$ is integer feasible **then** $\mathsf{UB} \leftarrow z_{\mathsf{LP}}^{\mathsf{N}^*}$; $\mathbf{x}^* \leftarrow \mathbf{x}_{\mathsf{LP}}^{\mathsf{N}^*}$; Prune (remove) $\mathsf{N} \in \mathcal{L}$ with $z_{\mathsf{LP}}^{\mathsf{N}} \geq \mathsf{UB}$.
9:         **goto** Line 18
10:    **end if**
11:    Decide whether to add cutting planes (**goto** Line 12) or branch (**goto** Line 15).
12:    **Cut Selection:** Generate candidate cuts C; Select $\mathsf{C}^* \subseteq \mathsf{C}$ via cut selection policy.
13:    Add $\mathsf{C}^*$ to $\mathsf{N}^*$'s formulation. Keep $\mathsf{N}^*$ unchanged if $\mathsf{C}^* = \varnothing$.
14:    Update $\mathcal{L} \leftarrow \mathcal{L} \cup \{\mathsf{N}^*\}$, $\mathsf{LB} \leftarrow \min_{\mathsf{N} \in \mathcal{L}}\{z_{\mathsf{LP}}^{\mathsf{N}}\}$, and **goto** Line 18
15:    **Branching:** Select fractional variable index $j \in [n_1]$ from $\mathbf{x}_{\mathsf{LP}}^{\mathsf{N}^*}$ via branching policy.
16:    Create $\mathsf{N}_L, \mathsf{N}_R$ from $\mathsf{N}^*$ by adding constraints $\mathbf{x}_j \leq \left\lfloor \left(\mathbf{x}_{\mathsf{LP}}^{\mathsf{N}^*}\right)_j \right\rfloor$ and $\mathbf{x}_j \geq \left\lceil \left(\mathbf{x}_{\mathsf{LP}}^{\mathsf{N}^*}\right)_j \right\rceil$.
17:    Update $\mathcal{L} \leftarrow \mathcal{L} \cup \{\mathsf{N}_L, \mathsf{N}_R\}$, $\mathsf{LB} \leftarrow \min_{\mathsf{N} \in \mathcal{L}}\{z_{\mathsf{LP}}^{\mathsf{N}}\}$.
18:    $i \leftarrow i + 1$.
19: **end while**
**Ensure:** Best found incumbent solution $\mathbf{x}^*$ and final bounds $\mathsf{LB}, \mathsf{UB}$.

---

Heuristic procedures within B&C often rely on scoring rules to guide decisions. A prominent example arises in the crucial task of *cut selection* within state-of-the-art solvers like SCIP [Achterberg, 2009, Gamrath et al., 2020]. At a given state $s$ of the search (characterized, for instance, by the current problem $(A, \mathbf{b}, \mathbf{c})$, current LP solution $\mathbf{x}_{\mathsf{LP}}^*$, and potentially the best known integer feasible solution $\bar{\mathbf{x}}$, often called the incumbent solution), a set $\mathcal{A}^s$ of candidate cuts is available. A cut selection policy must then choose a beneficial subset from $\mathcal{A}^s$.

Many widely-used, human-designed heuristics implement this selection by scoring each candidate cut $a \in \mathcal{A}^s$, represented by $\boldsymbol{\alpha}^\mathsf{T} \mathbf{x} \leq \beta$. These policies employ various *feature extractors*, functions that map the pair $(s, a)$ to a real-valued quality measure. Key feature extractors used in SCIP include [Achterberg, 2007, Wesselmann and Stuhl, 2012, Gamrath et al., 2020]:

- Efficacy ($\phi_{\text{eff}}$): $\phi_{\text{eff}}(s, a) := (\boldsymbol{\alpha}^\mathsf{T} \mathbf{x}_{\mathsf{LP}}^* - \beta)/\|\boldsymbol{\alpha}\|_2$. This measures the Euclidean distance by which the current LP solution $\mathbf{x}_{\mathsf{LP}}^*$ violates the cut.

- Objective Parallelism ($\phi_{\text{obj}}$): $\phi_{\text{obj}}(s, a) := |\boldsymbol{\alpha}^\mathsf{T} \mathbf{c}|/(\|\boldsymbol{\alpha}\|_2 \|\mathbf{c}\|_2)$. This measures the similarity between the cut normal $\boldsymbol{\alpha}$ and the objective vector $\mathbf{c}$.

- Directed Cutoff Distance ($\phi_{\text{dcd}}$): This feature quantifies efficacy specifically in the direction from $\mathbf{x}_{\mathsf{LP}}^*$ towards the incumbent $\bar{\mathbf{x}}$, when $\bar{\mathbf{x}}$ is available and distinct from $\mathbf{x}_{\mathsf{LP}}^*$. It is computed as $\phi_{\text{dcd}}(s, a) := (\boldsymbol{\alpha}^\mathsf{T} \mathbf{x}_{\mathsf{LP}}^* - \beta)/(|\boldsymbol{\alpha}^\mathsf{T}(\bar{\mathbf{x}} - \mathbf{x}_{\mathsf{LP}}^*)|/\|\bar{\mathbf{x}} - \mathbf{x}_{\mathsf{LP}}^*\|_2)$.

- Integral Support ($\phi_{\text{int}}$): $\phi_{\text{int}}(s, a) := |\{j \in [n_1] \mid \boldsymbol{\alpha}_j \neq 0\}|/|\{j \in [n] \mid \boldsymbol{\alpha}_j \neq 0\}|$, where $n = n_1 + n_2$ is the total number of decision variables. This calculates the fraction of integer-constrained variables ($j \in [n_1]$) among all variables involved in the cut.

These feature extractors produce a vector $\phi(s, a) = (\phi_{\text{eff}}(s, a), \phi_{\text{obj}}(s, a), \phi_{\text{dcd}}(s, a), \phi_{\text{int}}(s, a))^\mathsf{T} \in \mathbb{R}^4$. Policies like SCIP's default Hybrid selector define the score using a linear function determined by a parameter vector $\mathbf{w} = (\mathbf{w}_1, \mathbf{w}_2, \mathbf{w}_3, \mathbf{w}_4)^\mathsf{T} \in \mathbb{R}^4$. The score is computed as:

$$f(s, a, \mathbf{w}) = \mathbf{w}^\mathsf{T} \phi(s, a) = \mathbf{w}_1 \phi_{\text{eff}}(s, a) + \mathbf{w}_2 \phi_{\text{obj}}(s, a) + \mathbf{w}_3 \phi_{\text{dcd}}(s, a) + \mathbf{w}_4 \phi_{\text{int}}(s, a).$$

Crucially, the choice of the weight vector $\mathbf{w}$ defines a specific cut selection policy within this linear scoring framework; different weights lead to different prioritizations of cuts. The weights $\mathbf{w}$ are

typically pre-tuned based on extensive computational experiments. SCIP's more advanced Ensemble selector also employs a weighted-sum score $f(s, a, \mathbf{w}') = (\mathbf{w}')^\mathsf{T}\phi'(s, a)$, but utilizes an extended feature vector $\phi'(s, a) \in \mathbb{R}^\ell$ (where $\ell$ may be larger than 4) that includes additional, often normalized, metrics like density, dynamism, pseudo-cost scores, and variable lock information [Achterberg, 2007]. The selection policy then chooses the cut $a^*$ that yields the highest score.

This practice of using parameterized, score-based rules is widespread in B&C. For instance, beyond the linear scoring approach exemplified by SCIP's Hybrid cut selector, other critical decisions also rely on similar mechanisms. Branching variable selection frequently employs heuristic scores derived from informative features, such as pseudo-costs which estimate the impact of branching on candidate variables [Achterberg, 2007, Achterberg et al., 2005]. Node selection strategies also commonly utilize scoring based on various node attributes [He et al., 2014, Yilmaz and Yorke-Smith, 2021]. The prevalence of such score-guided heuristics across multiple B&C components motivates our study of such policies within the following general framework.

Let $\mathcal{S}$ be the state space. Each state $s \in \mathcal{S}$ contains the information available at a given step of the branch-and-cut algorithm (Algorithm 1), including the tuple $(\mathcal{L}, \mathbf{x}^*, i, \mathsf{N}^*)$. Let $\mathcal{A}_k$ be the space of possible actions for a specific decision type $k$ (e.g., $k = 1$ for node selection, $k = 2$ for cut selection, $k = 3$ for branching). For a given state $s \in \mathcal{S}$, let $\mathcal{A}_k^s \subseteq \mathcal{A}_k$ denote the set of available actions of type $k$ for state $s$ (e.g., $\mathcal{A}_2^s = \mathcal{A}^s$ in the example above contains candidate cuts). The goal is to learn a parameterized scoring function $f_k : \mathcal{S} \times \mathcal{A}_k \times \mathcal{W}_k \to \mathbb{R}_+$, parameterized by $\mathbf{w}^k \in \mathcal{W}_k$, where $\mathcal{W}_k$ is the parameter space (e.g., the weight vector $\mathbf{w}$ in the SCIP Hybrid example belongs to $\mathcal{W}_2$). This function evaluates potential actions $a \in \mathcal{A}_k^s$. A fixed feature extractor function $\phi_k : \mathcal{S} \times \mathcal{A}_k \to \mathbb{R}^{\ell_k}$ maps the state-action pair $(s, a)$ to a feature vector (e.g., $\phi(s, a) \in \mathbb{R}^4$ for $k = 2$ in the example above). The scoring function often takes the form $f_k(s, a, \mathbf{w}^k) = \psi_k(\phi_k(s, a), \mathbf{w}^k)$, where $\psi_k$ could be an ML model like a Multi-Layer Perceptron (MLP, Definition 3.6) or, as exemplified by SCIP's Hybrid selector, a linear function $\psi_k(\phi_k(s, a), \mathbf{w}^k) = (\mathbf{w}^k)^\mathsf{T}\phi_k(s, a)$. The chosen action $a^*$ is typically one that maximizes the score, selected according to $a^* \in \arg\max_{a \in \mathcal{A}_k^s} f_k(s, a, \mathbf{w}^k)$

(with a consistent tie-breaking rule). Training the parameters $\mathbf{w}^k$ often relies on supervisory signals from an expert function or oracle, which provides high-quality action evaluations (e.g., Strong Branching scores [Alvarez et al., 2017], exact bound improvements from cuts [Paulus et al., 2022]) or assesses the overall impact on the search (e.g., reduction in runtime or tree size [Huang et al., 2022]). Supervised learning methods then use these expert-derived values as targets or labels to optimize $\mathbf{w}^k$.

## 2 Prior work and our contributions

We first discuss prior computational and theoretical work on using ML techniques to make B&C decisions, and then describe our contributions in that context. We will use the language of the general framework described above to discuss everything in a unified way.

### 2.1 Related empirical work

**Learning to select node.** (Action type $k = 1$) Selecting which node to explore next from the queue $\mathcal{L}$ of active nodes (i.e., the set of potential actions $\mathcal{A}_1^s = \mathcal{L}$) is a critical B&C decision. He et al. [2014] learn a linear scoring function where $f_1(s, a, \mathbf{w}^1) = (\mathbf{w}^1)^\mathsf{T}\phi_1(s, a)$, to score each candidate node $a \in \mathcal{A}_1^s$. For each candidate node $a$, its features $\phi_1(s, a)$, categorized as node, branching, and tree features, are obtained using information typically recorded by solvers. The node with the highest score is selected, with the policy trained to imitate an oracle that exclusively explores nodes leading to an optimal integer solution. Differing in scope, Yilmaz and Yorke-Smith [2021] learn a policy for child selection. Their scoring function uses an MLP classifier $\psi_1$. This MLP takes as input features comprising 29 base metrics (listed in their Table 1). These include features of the branched variable (e.g., its simplex basis status), features of the newly created child nodes (e.g., their bounds), and tree features (e.g., global bounds and current depth). The MLP computes scores $f_1(s, a, \mathbf{w}^1)$ for three distinct actions following a branch—exploring the left child node, the right child node, or both children, and selects the action with the highest score. Their expert policy is derived from paths to the top-$k$ known solutions.

**Learning to cut.** (Action type $k = 2$) In state $s$, an action $a \in \mathcal{A}_2^s$ corresponds to selecting candidate cuts at node $\mathsf{N}^*$. Huang et al. [2022] use fixed features $\phi_2(s, a) \in \mathbb{R}^{14}$ (detailed in their

Table 1) as input to an MLP $\psi_2$. This MLP outputs a score $f_2(s, a, \mathbf{w}^2) = \psi_2(\phi_2(s, a), \mathbf{w}^2)$ for each cut, and cuts with the highest scores are selected. Their expert oracle assesses the quality of sets (bags) of cuts based on overall runtime reduction. Paulus et al. [2022] also utilize a fixed initial feature representation (as detailed in their Table 5, based on Gasse et al. [2019]), encoding the LP relaxation and candidate cuts $\mathcal{A}_2^s$ as a tripartite graph. Their scoring function is a complex model comprising a Graph Convolutional Neural Network (GCNN), an attention mechanism, and a final MLP (the entire set of parameters for this composite model is learned from data). This function produces a score for each cut, and cuts with the highest scores are then selected. Their expert signal is the exact LP bound improvement (Lookahead score) from individual cuts.

**Learning to branch.** (Action type $k = 3$) In state $s$, an action $a \in \mathcal{A}_3^s$ represents choosing a fractional variable $a \in [n_1]$ to branch on within node $\mathsf{N}^*$. Strong Branching (SB) often serves as the expert oracle for this task. Khalil et al. [2016] proposed a linear scoring function $f_3(s, a, \mathbf{w}^3) = (\mathbf{w}^3)^\top \phi_3(s, a)$. Their fixed feature extractor $\phi_3(s, a)$ computes features (based on 72 atomic features as detailed in their Table 1) describing candidate variable $a$ in the context of the current node. The variable $a$ with the highest score $f_3(s, a, \mathbf{w}^3)$ is selected for branching. Similarly, Alvarez et al. [2017] used an Extremely Randomized Trees model [Geurts et al., 2006] to directly predict SB scores $f_3(s, a, \mathbf{w}^3)$; here, the parameters $\mathbf{w}^3$ define the learned structure of the tree ensemble. Their fixed feature extractor $\phi_3(s, a)$ calculates features for variable $a$ based on static problem data, dynamic information from the current LP solution, and optimization history. The variable $a \in [n_1]$ yielding the highest predicted SB score is chosen.

These learning-based strategies empirically demonstrate the potential to automate the design of high-performance policies for various decisions within B&C solvers, including node selection, cut selection, and branching, by leveraging data and expert knowledge in a structured manner. Consequently, establishing a rigorous theoretical foundation for these learned policies, particularly regarding their sample complexity and generalization guarantees, becomes essential.

## 2.2 Related theoretical work

In statistical learning theory, the goal is to find a parameter $\mathbf{w} \in \mathcal{W}$ that minimizes the expected cost $\mathbb{E}_{I \sim \mathcal{D}}[V(I, \mathbf{w})]$, where $V : \mathcal{I} \times \mathcal{W} \to [0, H]$ is a cost function bounded above by some $H > 0$, and $\mathcal{D}$ is an unknown distribution over the instance space $\mathcal{I}$. In the context of branch-and-cut, $V(I, \mathbf{w})$ represents a measure of the overall performance (e.g., final tree size, solution time) when applying a policy (for node, cut, and branching variable selections) parameterized by $\mathbf{w}$ to solve the initial problem instance $I$. Since $\mathcal{D}$ is unknown, we usually rely on an empirical estimate derived from a finite sample $\{I_1, \ldots, I_N\} \subseteq \mathcal{I}$ drawn independently and identically distributed (i.i.d.) from $\mathcal{D}$, by selecting the parameter $\mathbf{w}$ that minimizes the empirical average $\frac{1}{N} \sum_{i=1}^N V(I_i, \mathbf{w})$. A fundamental question concerns the *sample complexity* of uniform convergence: how fast does the empirical average converge to the true expectation, uniformly for all parameters $\mathbf{w} \in \mathcal{W}$, as the sample size $N$ increases?

Standard results relate this uniform convergence rate to measures of the complexity of the function class $\mathcal{V} = \{V(\cdot, \mathbf{w}) : \mathcal{I} \to [0, H] \mid \mathbf{w} \in \mathcal{W}\}$. One such measure is the pseudo-dimension, $\mathrm{Pdim}(\mathcal{V})$, and classical results (see, e.g., Pollard [1984]) give bounds of the form: with probability at least $1 - \delta$,

$$\sup_{\mathbf{w} \in \mathcal{W}} \left| \frac{1}{N} \sum_{i=1}^N V(I_i, \mathbf{w}) - \mathbb{E}_{I \sim \mathcal{D}}[V(I, \mathbf{w})] \right| = \mathcal{O}\left( H \sqrt{\frac{\mathrm{Pdim}(\mathcal{V}) + \log(1/\delta)}{N}} \right). \tag{1}$$

A significant line of research establishing rigorous sample complexity bounds for data-driven algorithm configuration was initiated by Gupta and Roughgarden [2016]. This PAC-style approach inspired investigations into the learnability of parameters for various algorithms by analyzing the structure of algorithm performance with respect to its parameters. Recent applications include configuring components within B&C (such as variable selection [Balcan et al., 2024b] and cut generation/selection [Balcan et al., 2021c, 2022b, 2021a, Cheng et al., 2024, Cheng and Basu, 2024]), learning heuristic functions for graph search algorithms [Sakaue and Oki, 2022], tuning parameters for iterative methods like gradient descent [Jiao et al., 2025], configuring data-driven projections for linear programming [Sakaue and Oki, 2024], analyzing combinatorial partitioning problems [Balcan et al., 2017], mechanism design for revenue maximization [Balcan et al., 2025b], tuning ElasticNet

[Balcan et al., 2022a], hyperparameter tuning in neural networks Balcan et al. [2025a], and learning decision trees [Balcan and Sharma, 2024]. Building on the techniques used across these diverse applications, Balcan et al. formalized and generalized the approach, particularly emphasizing the role of the *piecewise structure* of algorithm performance in establishing sample complexity guarantees [Balcan et al., 2024a,c]. Related theoretical foundations, surveys, and analyses are also available in [Balcan, 2020, Balcan et al., 2021b].

The first application of this PAC learning paradigm specifically to the B&C setting was presented by Balcan et al. [2024b] (first appearing in ICML 2018), who analyzed learning *linear variable selection* policies. They showed that the B&C tree size is a piecewise constant function of the linear weights, enabling pseudo-dimension bounds. Subsequently, Balcan et al. [2021c] extended this line of research significantly. They considered learning parameters for generating Chvátal-Gomory (CG) cuts and policies selecting cuts based on weighted combinations of standard scoring rules. They showed that sequential CG cuts induce a piecewise structure defined by multivariate polynomials, and that weighted scoring rules lead to piecewise constant behavior with respect to the weights. Importantly, Balcan et al. [2021c] also introduced and analyzed a *general model of tree search*, providing sample complexity bounds for simultaneously tuning multiple components (like node, variable, and cut selection) when guided by *linear scoring functions*. Further refinements came from Balcan et al. [2021a], who exploited the *path-wise* nature of many common B&C scoring rules within their general tree search model to derive exponentially sharper bounds, dependent on tree depth rather than total nodes, for policies using such rules. Addressing the challenge of infinite cut families, Balcan et al. [2022b] analyzed the learnability of Gomory Mixed-Integer (GMI) cut parameters, establishing a piecewise structure defined by degree-10 polynomial hypersurfaces.

More recently, theoretical work has begun to incorporate nonlinear models used in practice. Cheng et al. [2024] studied the sample complexity of using neural networks to map instances directly to algorithm parameters (specifically for cut selection), analyzing the complexity in terms of the neural network weights. Expanding on the types of cuts considered, Cheng and Basu [2024] investigated learning parameters for general *Cut Generating Functions (CGFs)*. CGFs provide a broad theoretical framework for deriving cutting planes, generalizing classical families like CG and GMI cuts (see [Basu et al., 2015] for a survey), and thus represent a promising avenue for enhancing B&C performance. Cheng and Basu [2024] derived sample complexity bounds for learning parameters within specific CGF families and also explored learning instance-dependent CGF parameters using neural networks.

## 2.3  Our contribution

The present paper contributes to this theoretical foundation by establishing rigorous sample complexity bounds for learning B&C policies where the scoring functions $f_k(s, a, \cdot)$ guiding each decision step (node/cut/branching variable selection) belong to the class of *piecewise polynomial functions* with respect to the parameters $\mathbf{w}$. This class, formalized in Definition 3.1, provides a unifying lens. It naturally includes the *linear* models studied previously [Balcan et al., 2024b, 2021c] and, as shown below in Proposition 3.4, our results improve upon the existing bounds for this case [Balcan et al., 2021c, Theorem 5.2]. Crucially, this framework also encompasses the *neural network* architectures (e.g., MLPs with ReLU activations) prevalent in recent work described above in Section 2.1. Lemma 3.7 below, building on Anthony and Bartlett [1999] and Bartlett et al. [2019], confirms that such networks yield scoring functions with the required piecewise polynomial structure.

By establishing that piecewise polynomial scoring functions induce a related piecewise structure in the overall cost metric $V(I, \cdot)$ (Theorem 3.3), our work bridges the gap between prior theory (largely focused on linear scoring rules [Balcan et al., 2021c,a, 2024b]) and current practice involving neural networks. We derive pseudo-dimension bounds (Propositions 3.4 and 3.8) applicable to both linear and neural network-based policies, leading to distribution-independent sample complexity guarantees via (1). The identified structure also opens the door to potentially tighter, data-dependent guarantees via empirical Rademacher complexity (Proposition 3.9).

Our techniques apply to more general sequential decision making problems beyond branch-and-cut. We develop this more general theory first in Section 3 and then derive the consequences for branch-and-cut in Section 4.

# 3 General Framework and Sample Complexity Bounds

We consider a general setting involving an iterative procedure operating on a state $s$ from a state space $\mathcal{S}$. The procedure starts from an initial state $s_0 \in \mathcal{S}$, which is typically derived from an initial problem instance $I \in \mathcal{I}$. The procedure may consist of multiple rounds (e.g., corresponding to processing nodes in a B&C tree), where each round involves executing $d$ distinct types of actions sequentially. For any state $s \in \mathcal{S}$, there exists a set $\mathcal{A}_k^s$ of available actions of action type $k \in [d]$. A transition function determines the next state $s'$ if action $a \in \mathcal{A}_k^s$ is selected in state $s$. The selection of an action $a^* \in \mathcal{A}_k^s$ is guided by a parameterized scoring function $f_k : \mathcal{S} \times \mathcal{A}_k \times \mathcal{W}_k \to \mathbb{R}_+$, where $\mathcal{A}_k = \bigcup_{s \in \mathcal{S}} \mathcal{A}_k^s$. The function $f_k$ takes the current state $s$, a candidate action $a$, and a parameter vector $\mathbf{w}^k \in \mathcal{W}_k \subseteq \mathbb{R}^{W_k}$ as input and returns a score. Fixing the parameters $\mathbf{w}^1, \ldots, \mathbf{w}^d$ determines a decision policy, and the goal is to select a good policy, i.e., a good set of parameters, as determined by penalty functions $P_k : \mathcal{S} \times \mathcal{A}_k \times \mathbb{N} \to \mathbb{R}$ for each action type $k$. The value $P_k(s, a, i)$ represents the penalty obtained when taking action $a$ in state $s$ and round $i$. Algorithm 2 outlines this general process, including penalty accumulation, employing a greedy action selection strategy based on the scoring functions at each step.

---

**Algorithm 2** A Sequential Decision Process that Generalizes B&C

---

**Require:** Initial state $s_0 \in \mathcal{S}$, terminal states $\bar{\mathcal{S}} \subseteq \mathcal{S}$, policy parameters $\mathbf{w}^k \in \mathcal{W}_k$, scoring functions $f_k : \mathcal{S} \times \mathcal{A}_k \times \mathcal{W}_k \to \mathbb{R}$, penalty functions $P_k : \mathcal{S} \times \mathcal{A}_k \times \mathbb{N} \to \mathbb{R}$ for $k \in [d]$, max rounds $M$.
1: Initialize $s \leftarrow s_0$, $i \leftarrow 0$, $V \leftarrow 0$.
2: **while** $s \notin \bar{\mathcal{S}}$ and $i < M$ **do**
3:     **for** $k = 1$ to $d$ **do**
4:         Determine available actions $\mathcal{A}_k^s$ for state $s$.
5:         **if** $\mathcal{A}_k^s = \emptyset$ **then continue**
6:         **end if**
7:         Select action $a^* \leftarrow \arg\max_{a \in \mathcal{A}_k^s} f_k(s, a, \mathbf{w}^k)$. Break ties by lexicographic order.
8:         Accumulate penalty $V \leftarrow V + P_k(s, a^*, i)$.
9:         Compute the state $s'$ that results from applying action $a^*$ in state $s$.
10:       Update state $s \leftarrow s'$.
11:       **if** $s \in \bar{\mathcal{S}}$ **then break**
12:       **end if**
13:     **end for**
14:     $i \leftarrow i + 1$.
15: **end while**
**Ensure:** $s$ and $V$.

---

## 3.1 Worst-case sample complexity bounds

Let $V(I, \mathbf{w})$ denote the total penalty (overall cost) obtained when Algorithm 2 is executed starting from an initial state derived from instance $I \in \mathcal{I}$, using a policy parameterized by $\mathbf{w} = (\mathbf{w}^1, \ldots, \mathbf{w}^d)$, where each component $\mathbf{w}^k \in \mathcal{W}_k \subseteq \mathbb{R}^{W_k}$ provides the parameters for the $k$-th action type's scoring function. The overall policy parameter space is $\mathcal{W} = \prod_{k=1}^d \mathcal{W}_k \subseteq \mathbb{R}^W$, with $W = \sum_{k=1}^d W_k$ being the total number of parameters. This overall cost $V(I, \mathbf{w})$ serves as a measure of the policy's effectiveness (e.g., runtime), and the goal is to find $\mathbf{w}$ that minimizes $\mathbb{E}_{I \sim \mathcal{D}}[V(I, \mathbf{w})]$ (see Section 2). To analyze the sample complexity of learning $\mathbf{w}$ via i.i.d samples from $\mathcal{D}$, we aim to bound the complexity of the function class $\mathcal{V} = \{V(\cdot, \mathbf{w}) : \mathcal{I} \to [0, H] \mid \mathbf{w} \in \mathcal{W}\}$, typically via its pseudo-dimension $\text{Pdim}(\mathcal{V})$. Our theoretical analysis relies on specific structural properties of the scoring functions $f_k$ employed at each step of Algorithm 2. To formalize this, let us first define a general structural property that has a piecewise behavior.

**Definition 3.1.** Let $\mathcal{G}$ be a class of real-valued functions defined on a domain $\mathcal{X} \subseteq \mathbb{R}^\ell$ for some $\ell \in \mathbb{N}$. We say $\mathcal{G}$ has a $(\Gamma, \gamma, \beta)$-*structure* if for any finite collection of $N$ functions $g_1, \ldots, g_N \in \mathcal{G}$ with $N \geq \gamma$, the domain $\mathcal{X}$ can be partitioned into at most $N^\gamma \Gamma$ disjoint regions such that within each region, every function $g_j : \mathcal{X} \to \mathbb{R}$ ($j = 1, \ldots, N$) is a fixed polynomial of degree at most $\beta$.

Let $\mathcal{F}_k^* = \{f_k(s, a, \cdot) : \mathcal{W}_k \to \mathbb{R}_+ \mid (s, a) \in \mathcal{S} \times \mathcal{A}_k\}$ denote the class of functions, indexed by state-action pairs $(s, a)$, derived from the scoring functions for action type $k \in [d]$. We assume that

$\mathcal{F}_k^*$ exhibits such a $(\Gamma_k, \gamma_k, \beta_k)$-structure over the parameter space $\mathcal{W}_k$ for all $k \in [d]$. Furthermore, we assume the number of available actions for any state $s$ and action type $k$ is uniformly bounded: $|\mathcal{A}_k^s| \leq \rho_k$ for constants $\rho_k \geq 2$.

The focus on scoring functions arising from classes $\mathcal{F}_k^*$ possessing this $(\Gamma_k, \gamma_k, \beta_k)$-structure is well-founded for two main reasons:

1. This structure naturally includes linear policies ($f_k^{\mathrm{L}}(s, a, \mathbf{w}) = (\mathbf{w}^k)^T \phi_k(s, a)$ using fixed feature extractors $\phi_k$). For any fixed state-action pair $(s, a)$, the function $f_k^{\mathrm{L}}(s, a, \cdot)$ is a single polynomial (in $\mathbf{w}$) of degree $\beta = 1$ over the entire parameter space $\mathcal{W}$. Thus, the corresponding class $(\mathcal{F}_k^{\mathrm{L}})^*$ has a $(1, 0, 1)$-structure. Such linear scoring models were investigated in [Balcan et al., 2021c] (which generalizes [Balcan et al., 2024b]).

2. This structural assumption allows us to analyze modern deep learning practices. If scoring functions are implemented as MLPs with piecewise polynomial activations, the resulting function class $(\mathcal{F}_k^{\mathrm{MLP}})^*$ possesses a $(\Gamma_k, \gamma_k, \beta_k)$-structure. Lemma 3.7 formally establishes this, detailing the specific parameters $\Gamma_k, \gamma_k, \beta_k$ based on the MLP's characteristics like depth, width, degree of activation etc. This confirms our theoretical framework's applicability to these widely used modern models, aligning with empirical research discussed in Section 2.1.

The following lemma connects the pseudo-dimension of the function class $\mathcal{H} = \{h(\cdot, \mathbf{w}) : \mathcal{I} \to \mathbb{R} \mid \mathbf{w} \in \mathcal{W}\}$ to the structural properties of its dual class $\mathcal{H}^* = \{h(I, \cdot) : \mathcal{W} \to \mathbb{R} \mid I \in \mathcal{I}\}$, where $h : \mathcal{I} \times \mathcal{W} \to \mathbb{R}$ is a general function mapping instance-parameter pairs to outputs. Specifically, it bounds $\mathrm{Pdim}(\mathcal{H})$ based on the $(\Gamma, \gamma, \beta)$-structure of $\mathcal{H}^*$.

**Lemma 3.2.** Let $h : \mathcal{I} \times \mathcal{W} \to \mathbb{R}$, where $\mathcal{W} \subseteq \mathbb{R}^W$ for some $W \in \mathbb{N}_+$. If $\mathcal{H}^*$ has a $(\Gamma, \gamma, \beta)$-structure with $(\Gamma, \gamma, \beta) \in \mathbb{N}_+ \times \mathbb{N} \times \mathbb{N}$, then the pseudo-dimension of $\mathcal{H}$ satisfies:

$$\mathrm{Pdim}(\mathcal{H}) \leq 4 \left( \gamma \log(2\gamma + 1) + W \log(4e\beta + 1) + \log(2\Gamma) \right).$$

**Theorem 3.3.** Consider the sequential decision process defined in Algorithm 2. Assume each scoring function class $\mathcal{F}_k^*$ has a $(\Gamma_k, \gamma_k, \beta_k)$-structure with $(\Gamma_k, \gamma_k, \beta_k) \in \mathbb{N}_+ \times \mathbb{N} \times \mathbb{N}_+$, and $|\mathcal{A}_k^s| \leq \rho_k$. Let $\widetilde{\gamma} = \sum_{k=1}^d \gamma_k$, $\bar{\rho} = \prod_{k=1}^d \rho_k$, $\bar{\Gamma} = \prod_{k=1}^d \Gamma_k$, and recall that $W = \sum_{k=1}^d W_k$. Then, $\mathcal{V}^*$ has a $(\Gamma', \gamma', 0)$-structure with $\Gamma' = 2^d \bar{\rho}^{(\widetilde{\gamma}+W)(M+1)} \bar{\Gamma} \left( e \sum_{k=1}^d \rho_k^2 \beta_k / W \right)^W$ and $\gamma' = \widetilde{\gamma} + W$.

Theorem 3.3 establishes the crucial property that if the scoring function classes $\mathcal{F}_k^*$ possess a $(\Gamma_k, \gamma_k, \beta_k)$-structure, then the resulting cost function dual class $\mathcal{V}^*$ exhibits a $(\Gamma', \gamma', 0)$-structure, i.e., a piecewise constant structure. This result, when combined with Lemma 3.2, yields bounds on the pseudo-dimension $\mathrm{Pdim}(\mathcal{V})$. These bounds, in turn, enable the derivation of sample complexity guarantees through standard uniform convergence results, such as (1). See a more detailed discussion of the uniform convergence results in Appendix C.2.

As a specific application, consider the case of linear scoring functions:

**Proposition 3.4.** Let $\mathcal{V}^{\mathrm{L}}$ be the cost function class obtained when using linear scoring functions $f_k(s, a, \mathbf{w}^k) = (\mathbf{w}^k)^\mathsf{T} \phi_k(s, a)$ for all $k \in [d]$. Then, $\mathrm{Pdim}\left(\mathcal{V}^{\mathrm{L}}\right) = \mathcal{O}\left(WM \sum_{k=1}^d \log \rho_k\right)$.

**Remark 3.5.** Proposition 3.4 improves upon the bound established in Theorem 5.2 by Balcan et al. [2021c] for linear scoring functions: $\mathrm{Pdim}(\mathcal{V}^{\mathrm{L}}) = \mathcal{O}\left(WM \sum_{k=1}^d \log \rho_k + W \log W\right)$. This improvement stems from our analysis technique, which leverages the particular structure inherent in our problem setting, rather than relying on the theorems presented in Balcan et al. [2024a] (see a related discussion in Appendix E, Bartlett et al. [2022]).

Turning to nonlinear models, particularly neural networks, we first establish in Lemma 3.7 that scoring functions implemented via MLPs (with suitable activations) satisfy the required structural property. This allows us to apply our general framework (Theorem 3.3 and Lemma 3.2) to derive pseudo-dimension bounds, and thus sample complexity results, for policies modeled by MLPs.

**Definition 3.6.** A Multi Layer Perceptron (MLP) computes $\mathrm{MLP}(\mathbf{x}, \mathbf{w}) : \mathbb{R}^d \times \mathbb{R}^W \to \mathbb{R}^\ell$ via the composition $\mathrm{MLP}(\mathbf{x}, \mathbf{w}) = (T_{L,\mathbf{w}} \circ \sigma^{(\cdot)} \circ T_{L-1,\mathbf{w}} \circ \cdots \circ \sigma^{(\cdot)} \circ T_{1,\mathbf{w}})(\mathbf{x})$. Here $T_{i,\mathbf{w}}$ denote affine transformations parameterized by $\mathbf{w} \in \mathbb{R}^W$, and $\sigma^{(\cdot)}$ denotes the element-wise application of

the activation function $\sigma : \mathbb{R} \to \mathbb{R}$. The network has $L \geq 1$ layers, $U$ total neurons, and $W$ total parameters. We assume the activation $\sigma$ is piecewise polynomial: its domain $\mathbb{R}$ can be partitioned into at most $p$ disjoint intervals such that, within each interval, $\sigma$ is defined by a univariate polynomial of degree at most $\alpha$.

**Lemma 3.7.** Consider an MLP as defined in Definition 3.6, characterized by $W, L, U, p, \alpha$. Then, the class $\{\mathsf{MLP}(\mathbf{x}, \cdot) : \mathbb{R}^W \to \mathbb{R} \mid \mathbf{x} \in \mathbb{R}^d\}$ has a $\left(2^L \alpha^{L^2 W} \left(2epU/W\right)^{LW}, LW, L\alpha^L\right)$-structure.

**Proposition 3.8.** Let $\mathcal{V}^{\mathsf{MLP}}$ be the cost function class obtained when using MLP scoring functions $f_k(s, a, \mathbf{w}^k) = \mathsf{MLP}_k(\phi_k(s, a), \mathbf{w}^k)$ for all $k \in [d]$. Then,

$$\mathrm{Pdim}\left(\mathcal{V}^{\mathsf{MLP}}\right) = \mathcal{O}\left(\left(\sum_{k=1}^d L_k W_k\right)\left(M\sum_{k=1}^d \log \rho_k + \log\left(\sum_{k=1}^d p_k U_k\right)\right) + W \log\left(\sum_{k=1}^d \alpha_k^{L_k}\right) + \sum_{k=1}^d L_k^2 W_k \log \alpha_k\right).$$

Specifically, for ReLU MLPs, we have $\alpha_k = 1$ and $p_k = 2$ for all $k \in [d]$, leading to the bound:

$$\mathrm{Pdim}\left(\mathcal{V}^{\mathsf{ReLU}}\right) = \mathcal{O}\left(\left(\sum_{k=1}^d L_k W_k\right)\left(M\sum_{k=1}^d \log \rho_k + \log\left(\sum_{k=1}^d U_k\right)\right)\right).$$

### 3.2 Data-dependent sample complexity bounds

Alternatively, a uniform convergence result similar to (1) can be obtained using the *empirical Rademacher complexity*. Let $S_N = \{I_1, \ldots, I_N\}$ be the sample drawn i.i.d. from $\mathcal{D}$. The empirical Rademacher complexity is $\widehat{\mathcal{R}}_{S_N}(\mathcal{V}) = \mathbb{E}_{\boldsymbol{\sigma} \sim \{-1,1\}^N}[\sup_{\mathbf{w} \in \mathcal{W}} \frac{1}{N}\sum_{i=1}^N \boldsymbol{\sigma}_i V(I_i, \mathbf{w})]$. Standard results (e.g., see Theorem 26.5 in [Shalev-Shwartz and Ben-David, 2014] and Theorem 3.3 in [Mohri et al., 2018]) provide the following uniform convergence guarantee: with probability at least $1 - \delta$,

$$\sup_{\mathbf{w} \in \mathcal{W}} \left|\frac{1}{N}\sum_{i=1}^N V(I_i, \mathbf{w}) - \mathbb{E}_{I \sim \mathcal{D}}[V(I, \mathbf{w})]\right| = \mathcal{O}\left(\widehat{\mathcal{R}}_{S_N}(\mathcal{V}) + H\sqrt{\frac{\log(1/\delta)}{N}}\right). \qquad (2)$$

Since $\widehat{\mathcal{R}}_{S_N}(\mathcal{V})$ is computed directly on the sample $S_N$, this bound is inherently *data-dependent* and can sometimes provide tighter estimates than worst-case guarantees like (1), which rely solely on the distribution-independent pseudo-dimension $\mathrm{Pdim}(\mathcal{V})$. Importantly, bounding $\widehat{\mathcal{R}}_{S_N}(\mathcal{V})$ relies fundamentally on the structural properties of the dual class $\mathcal{V}^*$, just like the case of pseudo-dimension.

Let $Q_{M,k}(I)$ denote the total number of distinct state-action pairs $(s, a)$ of type $k \in [d]$ encountered when Algorithm 2 is executed on an initial state derived from instance $I$ within its first $M$ rounds. The proof of Theorem 3.3 uses a worst-case estimate for $Q_{M,k}(I)$, namely $\rho_k \bar{\rho}^M$ (Lemma B.1). However, the presence of terminal states in Algorithm 2 often results in the actual number of encountered state-action pairs, $Q_{M,k}(I)$, being substantially smaller than this worst-case bound. The empirical Rademacher complexity bound presented below directly incorporates the sum of these $Q_{M,k}(I_i)$ values from the sample $\{I_1, \ldots, I_N\}$. A more detailed discussion and comparison of the bounds derived from pseudo-dimension and empirical Rademacher complexity can be found in Appendix C.2.

**Proposition 3.9.** Under the same hypothesis as Theorem 3.3, for any $S_N = \{I_1, \ldots, I_N\}$ with $N \geq \widetilde{\gamma} + W$, we have

$$\widehat{\mathcal{R}}_{S_N}(\mathcal{V}) \leq H\sqrt{\frac{2}{N}\left(d + \sum_{k=1}^d \log \Gamma_k + (\widetilde{\gamma} + W)\log\left(\sum_{k=1}^d \sum_{i=1}^N Q_{M,k}(I_i)\right) + W\log\left(\frac{e\sum_{k=1}^d \rho_k \beta_k}{W}\right)\right)}.$$

## 4 Application to Branch-and-Cut

Our general sequential decision process (Algorithm 2) can model the B&C algorithm (Algorithm 1), as discussed in Section 1 and Section 2. Starting from an initial state $s_0$ derived from an MIP instance $I = (A, \mathbf{b}, \mathbf{c})$, B&C iteratively makes decisions for node selection ($k = 1$), cut selection ($k = 2$), and branching ($k = 3$), corresponding to the $d = 3$ action types in our framework. The parameters $\mathbf{w} = (\mathbf{w}^1, \mathbf{w}^2, \mathbf{w}^3)$ governing these scoring functions can be trained via methods like imitation learning [He et al., 2014, Paulus et al., 2022, Yilmaz and Yorke-Smith, 2021, Alvarez et al., 2017] or reinforcement learning [Tang et al., 2020] to tune the respective policies (see Section 2.1).

Furthermore, the accumulated penalty $V$ in Algorithm 2 can directly measure B&C performance. For instance, setting immediate penalties $P_1(s, a, i) = 0$, $P_2(s, a, i) = 1$, and $P_3(s, a, i) = 2$ for all $(s, a, i) \in \mathcal{S} \times \mathcal{A} \times \mathbb{N}$ makes the cost $V : \mathcal{I} \times \mathcal{W} \to [0, 3M]$ equal to the size of the B&C tree. Consequently, minimizing $V$ is minimizing the total number of explored nodes, which is a common measure of B&C algorithmic efficiency, although the total runtime is also affected by other factors such as node processing time [Linderoth and Savelsbergh, 1999].

Modern solvers often prioritize aggressive cut generation at the root node of the B&C tree [Contardo et al., 2023]. This strategy leverages the global validity of root cuts and their potential for significant early impact [Contardo et al., 2023]. Theoretical work also suggests that restricting cuts to the root can be optimal under certain conditions [Kazachkov et al., 2024]. Motivated by these considerations, we analyze learning a cut selection policy ($k = 2$) via a ReLU neural network parameterized by $\mathbf{w}^2$. We assume cuts are added only at the root node for $R$ rounds before branching begins, where $R$ is typically small. In each round $i \in [R]$, the policy uses scores $f_2(s, a, \mathbf{w}^2)$ from the ReLU network to select at most $\kappa$ cuts from a pool of candidates. We assume a uniform upper bound $r$ on the number of available candidate cuts (typically, $r = \mathcal{O}(m + \kappa R)$, where $m$ is the number of constraints). We use fixed, deterministic rules for node and branching variable selection (e.g., depth-first search and a product scoring rule, respectively). Proposition 3.8 then implies the following.

**Proposition 4.1.** Let $T^{\text{ReLU}}(I, \mathbf{w}^2)$ denote the resulting B&C tree size for instance $I$ under the setup as discussed above. Then,
$$\text{Pdim}\left(\{T^{\text{ReLU}}(\cdot, \mathbf{w}^2) : \mathcal{I} \to \mathbb{R} \mid \mathbf{w}^2 \in \mathcal{W}_2\}\right) = \mathcal{O}\left(L_2 W_2 \left(\kappa R \log r + \log U_2\right)\right).$$

Our analysis also extends to simultaneously learning policies for all three core B&C decisions ($k = 1, 2, 3$), each governed by a separate ReLU network with parameters $\mathbf{w}^k \in \mathcal{W}_k$. We assume the standard bounds on the action space sizes: $\rho_1 \leq M$ available nodes for selection, $\rho_2 = \mathcal{O}(m + M)$ candidate cuts, and $\rho_3 = n$ candidate branching variables. Plugging into Proposition 3.8, we obtain:

**Proposition 4.2.** Let $V(I, \mathbf{w})$ denote the tree size when using B&C policies based on ReLU networks with parameters $\mathbf{w} = (\mathbf{w}^1, \mathbf{w}^2, \mathbf{w}^3)$. For the corresponding class $\mathcal{V}^{\text{ReLU}}$, we have:
$$\text{Pdim}\left(\mathcal{V}^{\text{ReLU}}\right) = \mathcal{O}\left((L_1 W_1 + L_2 W_2 + L_3 W_3)\left(M\left(\log(m + M) + \log n\right) + \log\left(U_1 + U_2 + U_3\right)\right)\right).$$

## 5 Empirical Validation

To complement our theoretical analysis, we conduct an experiment to test for consistency with the convergence rate suggested by our theoretical upper bounds (Equations (1) and (2)).

**Setup.** We trained a cut-selection policy[1], implemented as an MLP, to imitate an expert oracle based on normalized LP objective-value improvement, yielding a fixed parameter vector $\widehat{\mathbf{w}}$. The network uses two inputs (cut efficacy and objective parallelism), has two hidden layers with ten ReLU neurons each, and a single output with a clipped ReLU activation to produce a score in $[0, 1]$. We evaluated this fixed policy on two sets of packing instances from [Tang et al., 2020] with $m$ constraints and $n$ variables, using the configurations ($m = 10, n = 20$) and ($m = 15, n = 30$). The performance metric, $V(I, \widehat{\mathbf{w}})$, is the B&C tree size obtained when our MLP, parameterized by $\widehat{\mathbf{w}}$, selects the 10 cuts with the highest scores at the root node.

**Methodology.** Given training and test instances $\{I_i^{\text{train}}\}_{i=1}^N$ and $\{I_j^{\text{test}}\}_{j=1}^N$, drawn i.i.d. from a distribution $\mathcal{D}$, we measure the empirical generalization gap using the quantity
$$f(N) = \left| \frac{1}{N} \sum_{i=1}^N V(I_i^{\text{train}}, \widehat{\mathbf{w}}) - \frac{1}{N} \sum_{j=1}^N V(I_j^{\text{test}}, \widehat{\mathbf{w}}) \right|.$$

The motivation for this is grounded in the uniform convergence bounds (Equations (1) and (2)). These bounds establish that the deviation of any $N$-sample average from the true expectation is bounded by some $\varepsilon_N = \mathcal{O}(1/\sqrt{N})$. Because this guarantee applies to the averages over both the training and test sets, it follows from the triangle inequality that their difference, $f(N)$, is bounded by $2\varepsilon_N$. Thus, $f(N)$ serves as a direct, empirically measurable proxy for the convergence rate. To ensure statistical stability, we report the average $f(N)$ over 30 independent trials for each $N \in [1, 100]$.

---

[1] Our code is available at `https://github.com/Hongyu-Cheng/MLP4ScoreBnC`. The experiments were performed on a desktop computer with an Intel i7-12700F CPU (12 cores, 20 threads) and 32GB of RAM.

**Results.** To test whether the empirical data follows the $\mathcal{O}(1/\sqrt{N})$ convergence rate suggested by our theoretical upper bound, we fit the empirical data to the curve $g(N; a, b) = a/\sqrt{N} + b$. Figure 1 presents the results for both instance sets. The fitted curve closely tracks the empirical data, achieving high coefficients of determination ($R^2 = 0.951$ and $R^2 = 0.959$). This provides strong evidence that the generalization error converges in a manner consistent with our theoretical bound. Moreover, the result suggests that the constants in the generalization bounds are of a practical magnitude for this distribution of integer programs, bridging the gap between our theory and its application.

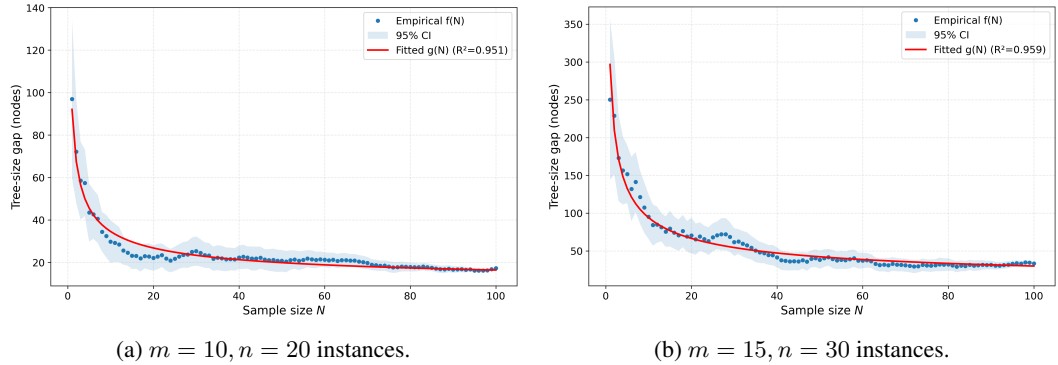

(a) $m = 10, n = 20$ instances.    (b) $m = 15, n = 30$ instances.

Figure 1: Empirical generalization gap $f(N)$ versus sample size $N$. The blue dots represent the average gap over 30 trials, the shaded region is the 95% confidence interval, and the red line is the fitted curve $g(N; a, b) = a/\sqrt{N} + b$.

## 6 Conclusions and Future Work

This paper establishes a theoretical framework for analyzing the generalization guarantees of learning policies within B&C and, more generally, within sequential decision-making processes. We demonstrate that if the scoring functions that guide the decisions exhibit a piecewise polynomial structure with respect to their learnable parameters, then the overall performance metric (e.g., tree size for B&C) is a piecewise constant function of these parameters. This structural insight enables us to derive pseudo-dimension bounds and corresponding sample complexity guarantees for policies parameterized not only by traditional linear models but also, significantly, by neural networks with piecewise polynomial activations such as ReLU. Our results thereby help bridge the gap between theoretical analyses, which have predominantly focused on linear policy classes, and the increasingly prevalent use of nonlinear, data-driven heuristics in contemporary algorithm configuration, offering a unified perspective for understanding their generalization performance.

Future investigations could build upon this work in several promising directions. One key theoretical challenge is to explore the expressive power and limitations of such parameterized policies. In particular, we aim to characterize their ability to approximate optimal strategies for these decision making problems, and analyze the associated bias-variance trade-offs inherent in the learning phase. A related issue is the critical role of the expert function or oracle that provides training signals; we need a deeper theoretical understanding of how the choice and quality of this expert influences the performance of the learned policy, ideally culminating in provable guarantees regarding its proximity to true optimality. The scope of our current framework could also be expanded. For instance, adapting the analysis to accommodate structured infinite action spaces, such as those involved in generating cuts using CGFs [Basu et al., 2015], would enhance its practical utility for integer programming. Moreover, enriching the underlying sequential decision model to formally include stochasticity in state transitions would allow our sample complexity results to address a wider set of dynamic and less predictable algorithmic environments.

## Acknowledgments and Disclosure of Funding

Both authors gratefully acknowledge support from Air Force Office of Scientific Research (AFOSR) grant FA9550-25-1-0038. The first author also acknowledges support from the Johns Hopkins University Mathematical Institute for Data Science (MINDS) Fellowship and the Duncan Award.

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

# A Auxiliary Lemmas

**Lemma A.1.** Let $d$ be a positive integer, and let $a_1, \ldots, a_d, b_1, \ldots, b_d > 0$. The following inequalities hold:

$$(1) \quad \log a_1 \leq \frac{a_1}{b_1} + \log\left(\frac{b_1}{e}\right),$$

$$(2) \quad \prod_{k=1}^{d} a_k^{b_k} \leq \left(\frac{\sum_{k=1}^{d} a_k b_k}{\sum_{k=1}^{d} b_k}\right)^{\sum_{k=1}^{d} b_k},$$

$$(3) \quad \left(\frac{ea_1}{b_1}\right)^{b_1} < \left(\frac{ea_1}{b_2}\right)^{b_2}, \quad \text{if } a_1 \geq b_2 > b_1.$$

*Proof.* (1) This follows from the observation that $\log(a_1/b_1) \leq a_1/b_1 - 1$.

(2) Since $f(x) = \log x$ is concave on $(0, +\infty)$, Jensen's inequality gives

$$\log\left(\sum_{k=1}^{d} \frac{b_k}{\sum_{j=1}^{d} b_j} a_k\right) \geq \sum_{k=1}^{d} \frac{b_k}{\sum_{j=1}^{d} b_j} \log a_k = \frac{\log\left(\prod_{k=1}^{d} a_k^{b_k}\right)}{\sum_{j=1}^{d} b_j},$$

which implies that

$$\log\left(\prod_{k=1}^{d} a_k^{b_k}\right) \leq \left(\sum_{k=1}^{d} b_k\right) \log\left(\frac{\sum_{k=1}^{d} a_k b_k}{\sum_{k=1}^{d} b_k}\right) = \log\left(\left(\frac{\sum_{k=1}^{d} a_k b_k}{\sum_{k=1}^{d} b_k}\right)^{\sum_{k=1}^{d} b_k}\right).$$

The inequality then holds because $\log x$ is increasing on $(0, +\infty)$.

(3) It suffices to prove that $g(x) = x\left(\log(ea_1) - \log x\right)$ is increasing on $(0, a_1]$. Note that its derivative is $g'(x) = \left(\log(ea_1) - \log x\right) + x(-1/x) = \log(a_1) - \log(x)$, which is positive for $x \in (0, a_1)$. The result follows since $g(x) = \log((ea_1/x)^x)$ and $\log(\cdot)$ is increasing. $\square$

**Lemma A.2** (Theorem 8.3 in [Anthony and Bartlett, 1999]). Let $\xi_1, \ldots, \xi_N : \mathbb{R}^W \to \mathbb{R}$ be $N$ multivariate polynomials of degree at most $\beta$ with $N \geq W$ and $\beta \in \mathbb{N}$. Then

$$|\{(\operatorname{sgn}(\xi_1(\mathbf{w})), \ldots, \operatorname{sgn}(\xi_N(\mathbf{w}))) : \mathbf{w} \in \mathbb{R}^W\}| \leq 2\left(\frac{2eN\beta}{W}\right)^W, \qquad \text{if } \beta \geq 1,$$

$$|\{(\operatorname{sgn}(\xi_1(\mathbf{w})), \ldots, \operatorname{sgn}(\xi_N(\mathbf{w}))) : \mathbf{w} \in \mathbb{R}^W\}| = 1, \qquad \text{if } \beta = 0.$$

where $e$ is the base of the natural logarithm.

**Definition A.3** (Pseudo-dimension). Let $\mathcal{H} = \{h(\cdot, \mathbf{w}) : \mathcal{I} \to \mathbb{R} \mid \mathbf{w} \in \mathcal{W}\}$ be a class of real-valued functions defined on an input space $\mathcal{I}$, parameterized by $\mathbf{w} \in \mathcal{W}$. The *pseudo-dimension* of $\mathcal{H}$, denoted $\operatorname{Pdim}(\mathcal{H})$, is the largest integer $N$ for which there exist instances $\{I_1, \ldots, I_N\} \subseteq \mathcal{I}$ and real threshold values $y_1, \ldots, y_N \in \mathbb{R}$ such that the function class $\mathcal{H}$ realizes all $2^N$ possible sign patterns on these instances relative to the thresholds:

$$|\{(\operatorname{sgn}(h(I_1, \mathbf{w}) - y_1), \ldots, \operatorname{sgn}(h(I_N, \mathbf{w}) - y_N)) \mid \mathbf{w} \in \mathcal{W}\}| = 2^N.$$

If no such finite largest $N$ exists, then $\operatorname{Pdim}(\mathcal{H}) = +\infty$.

*Proof of Lemma 3.2.* Consider any set of $N$ instance-witness pairs $(I_1, y_1), \ldots, (I_N, y_N) \in \mathcal{I} \times \mathbb{R}$, where $N \in \mathbb{N}_+$ and $N \geq \gamma$. By the lemma's hypothesis, the parameter space $\mathcal{W}$ can be partitioned into $K \leq N^\gamma \Gamma$ disjoint regions $Q_1, \ldots, Q_K$. Within each region $Q_i$ ($i \in [K]$), for every instance $I_j$ ($j \in [N]$), the function $h(I_j, \cdot)$ coincides with a fixed polynomial $\eta_{ji}(\mathbf{w})$ of degree at most $\beta$ for all $\mathbf{w} \in Q_i$. Let $\Pi(N)$ be the total number of distinct sign patterns over $\mathcal{W}$:

$$\Pi(N) = |\{(\operatorname{sgn}(h(I_1, \mathbf{w}) - y_1), \ldots, \operatorname{sgn}(h(I_N, \mathbf{w}) - y_N)) \mid \mathbf{w} \in \mathcal{W}\}|.$$

We can bound $\Pi(N)$ by summing the number of patterns within each region:

$$\Pi(N) \leq \sum_{i=1}^{K} |\{(\mathrm{sgn}(h(I_1, \mathbf{w}) - y_1), \ldots, \mathrm{sgn}(h(I_N, \mathbf{w}) - y_N)) \mid \mathbf{w} \in Q_i\}|$$

$$= \sum_{i=1}^{K} |\{(\mathrm{sgn}(\eta_{1i}(\mathbf{w}) - y_1), \ldots, \mathrm{sgn}(\eta_{Ni}(\mathbf{w}) - y_N)) \mid \mathbf{w} \in Q_i\}|. \tag{3}$$

We consider two cases based on the degree $\beta$:

**Case 1: $\beta = 0$.** In this case, each $\eta_{ji}$ is a constant. Therefore, the sum in (3) is bounded by $K \leq N^\gamma \Gamma$. By Definition A.3, the pseudo-dimension $\mathrm{Pdim}(\mathcal{H})$ is the largest integer $N$ such that $2^N \leq \Pi(N)$, and upper bounded by the largest $N$ such that $2^N \leq N^\gamma \Gamma$. Taking logarithms yields:

$$N \log(2) \leq \log(N^\gamma \Gamma) = \gamma \log(N) + \log(\Gamma)$$

$$\leq \gamma \left( \frac{N}{e(\gamma + 1)} + \log \frac{e(\gamma + 1)}{e} \right) + \log(\Gamma)$$

$$\leq \frac{1}{e} N + \gamma \log(\gamma + 1) + \log \Gamma,$$

where the second inequality follows from the first inequality in Lemma A.1. Rearranging gives $\left( \log(2) - \frac{1}{e} \right) N \leq \gamma \log(\gamma + 1) + \log(\Gamma)$. Since $(\log(2) - 1/e)^{-1} < 4$, this implies

$$\mathrm{Pdim}(\mathcal{H}) \leq 4(\gamma \log(\gamma + 1) + \log \Gamma).$$

**Case 2: $\beta \geq 1$.** We consider any $N \geq W$. Within each region $Q_i$, the functions $\eta_{ji}(\mathbf{w}) - y_j$ are polynomials of degree at most $\beta$. Applying Lemma A.2 to bound the number of sign patterns for the $N$ polynomials within each region $Q_i$ gives:

$$|\{(\mathrm{sgn}(\eta_{1i}(\mathbf{w}) - y_1), \ldots, \mathrm{sgn}(\eta_{Ni}(\mathbf{w}) - y_N)) \mid \mathbf{w} \in Q_i\}| \leq 2 \left( \frac{2eN\beta}{W} \right)^W.$$

Substituting this into the sum in (3) and using $K \leq N^\gamma \Gamma$, we get:

$$\Pi(N) \leq K \cdot 2 \left( \frac{2eN\beta}{W} \right)^W \leq N^\gamma \Gamma \cdot 2 \left( \frac{2eN\beta}{W} \right)^W.$$

Similarly, the pseudo-dimension $\mathrm{Pdim}(\mathcal{H})$ is bounded by the largest $N$ such that

$$2^N \leq 2N^\gamma \Gamma \left( \frac{2eN\beta}{W} \right)^W.$$

Taking logarithms:

$$N \log(2) \leq \log(2) + \gamma \log(N) + \log(\Gamma) + W \log \left( \frac{2eN\beta}{W} \right)$$

$$\leq \log(2) + \gamma \left( \frac{N}{2e\gamma + 1} + \log \frac{2e\gamma + 1}{e} \right) + \log \Gamma + W \left( \frac{2eN\beta/W}{4e^2\beta} + \log \frac{4e^2\beta}{e} \right)$$

$$\leq \log(2) + \frac{1}{2e} N + \gamma \log(2\gamma + 1) + \frac{1}{2e} N + W \log(4e\beta) + \log \Gamma$$

$$= \frac{1}{e} N + \gamma \log(2\gamma + 1) + W \log(4e\beta) + \log(2\Gamma).$$

Rearranging gives $\left( \log(2) - \frac{1}{e} \right) N \leq \gamma \log(2\gamma + 1) + W \log(4e\beta) + \log(2\Gamma)$. This implies

$$\mathrm{Pdim}(\mathcal{H}) \leq 4 \left( \gamma \log(2\gamma + 1) + W \log(4e\beta) + \log(2\Gamma) \right).$$

Combining both cases yields the claimed bound:

$$\mathrm{Pdim}(\mathcal{H}) \leq 4 \left( \gamma \log(2\gamma + 1) + W \log(4e\beta + 1) + \log(2\Gamma) \right). \qquad \square$$

# B Proofs of the results from Section 3.1

**Lemma B.1.** Assuming $\rho_j \geq 2$ for all $j \in [d]$, then for all $M \geq 1$ and all $k \in [d]$, we have

$$Q_{M,k}(I) \leq \rho_k \bar{\rho}^M,$$

where $\bar{\rho} = \prod_{j=1}^d \rho_j$.

*Proof of Lemma B.1.* Recall that $Q_{M,k}(I)$ is the total number of distinct state-action pairs $(s, a)$ of type $k \in [d]$ encountered when Algorithm 2 is executed starting from an initial state derived from $I$ within its first $M$ rounds. In the first round, it is easy to verify that there are at most $\prod_{j=1}^{k-1} \rho_j$ states that are about to take a type-$k$ action for all $k \in [d]$. Therefore, $Q_{1,k}(I) \leq \rho_k \cdot \prod_{j=1}^{k-1} \rho_j = \prod_{j=1}^k \rho_j$. Thus,

$$
\begin{aligned}
Q_{M,k}(I) &\leq Q_{M-1,k}(I) + \rho_k \cdot \left( \prod_{j=1}^{k-1} \rho_j \right) \bar{\rho}^{M-1} \\
&\leq \left( \prod_{j=1}^k \rho_j \right) \left( \sum_{i=0}^{M-1} \bar{\rho}^i \right) \\
&= \left( \prod_{j=1}^k \rho_j \right) \cdot \left( \frac{\bar{\rho}^M - 1}{\bar{\rho} - 1} \right) \\
&\leq \left( \prod_{j=1}^k \rho_j \right) \left( \rho_k \bar{\rho}^{M-1} \right), \\
&\leq \rho_k \bar{\rho}^M,
\end{aligned}
$$

where the second last inequality holds since we assume $\rho_j \geq 2$ for all $j$. This establishes the claim for round $M \in \mathbb{N}_+$ and all $k \in [d]$. $\square$

**Lemma B.2.** Consider an arbitrary collection of $N$ instances $S_N = \{I_1, \ldots, I_N\}$, where $N \geq \sum_{k=1}^d (\gamma_k + W_k)$. Let $\widetilde{Q}_{M,k} = \sum_{i=1}^N Q_{M,k}(I_i)$ denote the total number of distinct state-action pairs of type $k \in [d]$ encountered across all instances in $S_N$ within the first $M$ rounds of Algorithm 2. Recall $\bar{\Gamma} = \prod_{k=1}^d \Gamma_k$. Then, the number of distinct output vectors satisfies:

$$r(S_N) := |\{(V(I_1, \mathbf{w}), \ldots, V(I_N, \mathbf{w})) \mid \mathbf{w} \in \mathcal{W}\}| \leq 2^d \bar{\Gamma} \left( \prod_{k=1}^d \widetilde{Q}_{M,k}^{\gamma_k} \right) \left( \frac{e \sum_{k=1}^d \widetilde{Q}_{M,k} \rho_k \beta_k}{W} \right)^W.$$

*Proof of Lemma B.2.* The set of instances $S_N = \{I_1, \ldots, I_N\}$ define at most $N$ distinct initial states for Algorithm 2. Our objective is to prove that the parameter space $\mathcal{W} = \prod_{k=1}^d \mathcal{W}_k$ can be partitioned into at most $r(S_N)$ disjoint regions, such that within any given region, the execution sequence of Algorithm 2 (specifically, the sequence of states visited and actions taken) remains identical for all instances $I_1, \ldots, I_N$ when parameterized by any $\mathbf{w}$ from that region. Consequently, the final accumulated penalty $V(I_i, \mathbf{w})$ will be constant for each $i \in [N]$ within such a region.

Let us fix an action type $k \in [d]$. The total number of distinct type-$k$ state-action pairs encountered across all $N$ instances is bounded by the sum $\sum_{i=1}^N Q_{M,k}(I_i) = \widetilde{Q}_{M,k}$. The assumption that the function class $\mathcal{F}_k^* = \{f_k(s, a, \cdot) : \mathcal{W}_k \to \mathbb{R}_+ \mid (s, a) \in \mathcal{S} \times \mathcal{A}_k\}$ has a $(\Gamma_k, \gamma_k, \beta_k)$-structure allows us to apply Definition 3.1 to the collection of functions $\{f_k(s, a, \cdot)\}$ corresponding to all distinct state-action pairs encountered across the $N$ instances (whose total number is at most $\widetilde{Q}_{M,k} \geq N \geq \gamma_k$ by assumption). This application partitions the parameter space $\mathcal{W}_k$ into a collection of at most $(\widetilde{Q}_{M,k})^{\gamma_k} \Gamma_k$ disjoint regions. Within each such region resulting from this initial partition, every function $f_k(s, a, \cdot)$ is a fixed polynomial in $\mathbf{w}^k \in \mathcal{W}_k$ of degree at most $\beta_k$.

Now, consider any one fixed region obtained from this initial partition of $\mathcal{W}_k$. For the action selection $a^* \leftarrow \arg\max_{a \in \mathcal{A}_k^s} f_k(s, a, \mathbf{w}^k)$ to yield a consistent result for all $\mathbf{w}^k$ within this fixed region and for all relevant states $s$, the set of maximizers must be invariant. This invariance is ensured if the signs of the polynomial differences $\xi_{s,ij}(\mathbf{w}^k) := f_k(s, a^i, \mathbf{w}^k) - f_k(s, a^j, \mathbf{w}^k)$ are constant for all distinct pairs $a^i, a^j \in \mathcal{A}_k^s$ and all relevant states $s \in \Delta_k$. Here, $\Delta_k$ denotes the set of all states, across the instances, that are encountered during the $M$ rounds of executing Algorithm 2 and at which a type-$k$ action is about to be taken. Since $f_k(s, a^i, \cdot)$ and $f_k(s, a^j, \cdot)$ are fixed polynomials of degree at most $\beta_k$ in the current region, their difference $\xi_{s,ij}(\mathbf{w}^k)$ is also a polynomial of degree at most $\beta_k$. Consider the collection of all distinct polynomial differences $\{\xi_{s,ij}(\cdot)\}$ arising from all such states $s$ and all distinct pairs $a^i, a^j \in \mathcal{A}_k^s$. For each state $s$, there are $\binom{|\mathcal{A}_k^s|}{2}$ distinct pairs $\{a^i, a^j\}$. Since $|\mathcal{A}_k^s| \le \rho_k$, we have $\binom{|\mathcal{A}_k^s|}{2} \le |\mathcal{A}_k^s|\rho_k/2$. Thus, the total number of polynomial differences in the collection is bounded by

$$\sum_{s \in \Delta_k} \binom{|\mathcal{A}_k^s|}{2} \le \frac{\rho_k}{2} \sum_{s \in \Delta_k} |\mathcal{A}_k^s| \le \frac{\widetilde{Q}_{M,k}\rho_k}{2}.$$

The signs of these polynomial differences induce a refinement of the current fixed region into a finite number of subregions. According to Lemma A.2, the number of such subregions is at most

$$2\left(\frac{2e(\widetilde{Q}_{M,k}\rho_k/2)\beta_k}{W_k}\right)^{W_k}.$$

Within any single subregion generated by this refinement, the sign of every relevant polynomial difference $\xi_{s,ij}(\mathbf{w}^k)$ is constant for all $\mathbf{w}^k$ in that subregion. This constancy of signs implies that for any relevant state $s$, the outcome of the comparison between $f_k(s, a^i, \mathbf{w}^k)$ and $f_k(s, a^j, \mathbf{w}^k)$ (i.e., greater than, less than, or equal to) is fixed for all $\mathbf{w}^k$ in the subregion. Consequently, the set of actions achieving the maximum score, $\arg\max_{a \in \mathcal{A}_k^s} f_k(s, a, \mathbf{w}^k)$, is invariant throughout the subregion.

By applying this sign-based refinement to every region from the initial partition of $\mathcal{W}_k$, we obtain a final partition of $\mathcal{W}_k$. The total number of regions $r_k(S_N)$ in this final partition is bounded by the product of the number of initial regions and the maximum number of subregions per initial region:

$$r_k(S_N) \le \widetilde{Q}_{M,k}^{\gamma_k}\Gamma_k \cdot 2\left(\frac{2e(\widetilde{Q}_{M,k}\rho_k/2)\beta_k}{W_k}\right)^{W_k} = \widetilde{Q}_{M,k}^{\gamma_k}\Gamma_k \cdot 2\left(\frac{e\widetilde{Q}_{M,k}\rho_k\beta_k}{W_k}\right)^{W_k}.$$

The overall parameter space $\mathcal{W} = \prod_{k=1}^d \mathcal{W}_k$ is then partitioned by the Cartesian product of these final partitions for each $\mathcal{W}_k$. The total number of resulting regions $r(S_N)$ in $\mathcal{W}$ satisfies

$$r(S_N) \le \prod_{k=1}^d r_k(S_N) \le 2^d \prod_{k=1}^d \Gamma_k \cdot \prod_{k=1}^d \widetilde{Q}_{M,k}^{\gamma_k} \cdot \prod_{k=1}^d \left(\frac{e\widetilde{Q}_{M,k}\rho_k\beta_k}{W_k}\right)^{W_k}, \qquad (4)$$

Applying the second inequality from Lemma A.1, we bound the product term:

$$\prod_{k=1}^d \left(\frac{e\widetilde{Q}_{M,k}\rho_k\beta_k}{W_k}\right)^{W_k} \le \left(\frac{\sum_{k=1}^d W_k \cdot (e\widetilde{Q}_{M,k}\rho_k\beta_k/W_k)}{\sum_{k=1}^d W_k}\right)^{\sum_{k=1}^d W_k}$$

$$= \left(\frac{e\sum_{k=1}^d \widetilde{Q}_{M,k}\rho_k\beta_k}{W}\right)^W,$$

where $W = \sum_{k=1}^d W_k$. Substituting this back gives the final bound on the total number of regions in $\mathcal{W}$:

$$r(S_N) \le \prod_{k=1}^d \widetilde{Q}_{M,k}^{\gamma_k} \cdot 2^d \prod_{k=1}^d \Gamma_k \cdot \left(\frac{e\sum_{k=1}^d \widetilde{Q}_{M,k}\rho_k\beta_k}{W}\right)^W. \qquad (5)$$

Within each of these $r(S_N)$ disjoint regions of $\mathcal{W}$, the set $\arg\max_{a \in \mathcal{A}_k^s} f_k(s, a, \mathbf{w}^k)$ is invariant for all relevant decision steps. Assuming a consistent tie-breaking rule (e.g., lexicographical based on

action representation), the specific action $a^*$ chosen at each step is fixed throughout the region. Since this holds for all decision steps encountered during the execution of Algorithm 2 for any instance $I_i$ ($i \in [N]$), the entire sequence of states visited and actions taken is invariant for all $\mathbf{w}$ within the region. As the accumulated penalty $V(I_i, \mathbf{w})$ is solely determined by this fixed sequence, $V(I_i, \mathbf{w})$ must be constant as a function of $\mathbf{w}$ in the region. This invariance holds for all $i \in [N]$. This completes the proof. $\square$

*Proof of Theorem 3.3.* In the worst case, by Lemma B.1, we have the following holds for all $k \in [d]$,

$$\widetilde{Q}_{M,k} = \sum_{i=1}^{N} Q_{M,k}(I_i) \le N\rho_k\bar{\rho}^M,$$

where $\bar{\rho} = \prod_{k=1}^{d}\rho_k$. Plug this into the inequality (5) from Lemma B.2, we obtain that for $N \ge \sum_{k=1}^{d}(\gamma_k + W_k)$:

$$|\{(V(I_1, \mathbf{w}), \ldots, V(I_N, \mathbf{w})) \mid \mathbf{w} \in \mathcal{W}\}|$$

$$\le \prod_{k=1}^{d}\left(N\rho_k\bar{\rho}^M\right)^{\gamma_k} \cdot 2^d \prod_{k=1}^{d}\Gamma_k \cdot \left(\frac{eN\bar{\rho}^M \sum_{k=1}^{d}\rho_k^2\beta_k}{W}\right)^W$$

$$= N^{\widetilde{\gamma}}\bar{\rho}^{(M+1)\widetilde{\gamma}} \cdot 2^d \prod_{k=1}^{d}\Gamma_k \cdot N^W\bar{\rho}^{MW} \cdot \left(\frac{e\sum_{k=1}^{d}\rho_k^2\beta_k}{W}\right)^W$$

$$\le N^{\widetilde{\gamma}+W}\bar{\rho}^{(M+1)(\widetilde{\gamma}+W)} \cdot 2^d\bar{\Gamma} \cdot \left(\frac{e\sum_{k=1}^{d}\rho_k^2\beta_k}{W}\right)^W,$$

where $\widetilde{\gamma} = \sum_{k=1}^{d}\gamma_k$, $\bar{\Gamma} = \prod_{k=1}^{d}\Gamma_k$, and $W = \sum_{k=1}^{d}W_k$.

By Definition 3.1, this establishes that the function class $\mathcal{V}^* = \{V(I, \cdot) : \mathcal{W} \to \mathbb{R} \mid I \in \mathcal{I}\}$ has a $(\Gamma', \gamma', 0)$-structure, where $\gamma' = \widetilde{\gamma} + W = \sum_{k=1}^{d}(\gamma_k + W_k)$, which is no larger than $N$ as assumed in Lemma B.2, and

$$\Gamma' = 2^d\bar{\rho}^{(\widetilde{\gamma}+W)(M+1)}\bar{\Gamma}\left(\frac{e\sum_{k=1}^{d}\rho_k^2\beta_k}{W}\right)^W.$$

$\square$

*Proof of Proposition 3.4.* Direct verification confirms that the linear scoring function class $(\mathcal{F}_k^L)^*$ has a $(1, 0, 1)$-structure. In this case, the parameters from Theorem 3.3 specialize to $\widetilde{\gamma} = \sum_{k=1}^{d}\gamma_k = 0$, $\bar{\Gamma} = \prod_{k=1}^{d}\Gamma_k = 1$, and $\beta_k = 1$ for all $k$, which implies $\sum_{k=1}^{d}\rho_k^2\beta_k = \sum_{k=1}^{d}\rho_k^2$. Thus, the associated class $(\mathcal{V}^L)^*$ possesses a $(\Gamma', \gamma', 0)$-structure with $\gamma' = W$ and

$$\Gamma' = 2^d\bar{\rho}^{W(M+1)}\left(\frac{e\sum_{k=1}^{d}\rho_k^2}{W}\right)^W.$$

Applying the bound from Lemma 3.2 yields:

$$\text{Pdim}\left(\mathcal{V}^L\right)$$

$$\leq 4\left(W\log(2W+1) + \log(2\Gamma')\right)$$

$$= 4\left(W\log(2W+1) + (d+1)\log 2 + W(M+1)\sum_{k=1}^{d}\log\rho_k + W\log\left(e\sum_{k=1}^{d}\rho_k^2\right) - W\log W\right)$$

$$= 4\left(W\log\left(\frac{2W+1}{W}\right) + (d+1)\log 2 + W(M+1)\sum_{k=1}^{d}\log\rho_k + W\log\left(e\sum_{k=1}^{d}\rho_k^2\right)\right)$$

$$\leq 4\left(W\log(3e) + 2W\log\left(\prod_{k=1}^{d}\rho_k\right) + (d+1)\log 2 + W(M+1)\sum_{k=1}^{d}\log\rho_k\right)$$

$$= \mathcal{O}\left(WM\sum_{k=1}^{d}\log\rho_k\right).$$

$\square$

*Proof of Lemma 3.7.* Let $W_i$ locally denote the number of parameters in the first $i$ layers of the MLP, and let $U_i$ denote the number of neurons in the $i$-th layer, for $i \in [L]$. Following the proof technique of Theorem 7 in [Bartlett et al., 2019], for any $N$ inputs $\mathbf{x}_1, \ldots, \mathbf{x}_N$ with $N \geq LW$, the parameter space $\mathcal{W}$ can be partitioned into at most

$$2^L\left(\frac{2eNp\sum_{i=1}^{L}U_i\left(1+(i-1)\alpha^{i-1}\right)}{\sum_{i=1}^{L}W_i}\right)^{\sum_{i=1}^{L}W_i}$$

$$\leq 2^L\left(\frac{2eNp\sum_{i=1}^{L}U_iL\alpha^L}{\sum_{i=1}^{L}W_i}\right)^{\sum_{i=1}^{L}W_i}$$

$$= 2^L\left(\frac{2eNpUL\alpha^L}{\sum_{i=1}^{L}W_i}\right)^{\sum_{i=1}^{L}W_i}$$

$$\leq 2^L\left(\frac{2eNpUL\alpha^L}{LW}\right)^{LW}$$

$$= N^{LW}\cdot 2^L\alpha^{L^2W}\left(\frac{2epU}{W}\right)^{LW}$$

disjoint regions, where the last inequality is due to the third inequality in Lemma A.1 and the fact that $N \geq LW$. Within each region, the function $\text{MLP}(\mathbf{x}_j, \cdot) : \mathcal{W} \to \mathbb{R}$ is a fixed polynomial in $\mathbf{w}$ of degree at most $L\alpha^L$ for every $j \in [N]$.

This decomposition shows that the function class $\{\text{MLP}(\mathbf{x}, \cdot) : \mathcal{W} \to \mathbb{R} \mid \mathbf{x} \in \mathbb{R}^d\}$ has a $(\Gamma, \gamma, \beta)$-structure, where $\Gamma = 2^L\alpha^{L^2W}\left(\frac{2epU}{W}\right)^{LW}$, $\gamma = LW$, and $\beta = L\alpha^L$. $\square$

*Proof of Proposition 3.8.* According to Lemma 3.7, we have that

$$\Gamma_k = 2^{L_k}\alpha_k^{L_k^2W_k}\left(\frac{2ep_kU_k}{W_k}\right)^{L_kW_k}, \quad \gamma_k = L_kW_k, \quad \text{and} \quad \beta_k = L_k\alpha_k^{L_k}.$$

Let $\Lambda = \sum_{k=1}^{d} L_k W_k$, $\widetilde{L} = \sum_{k=1}^{d} L_k$. The parameters in Theorem 3.3 are $\widetilde{\gamma} = \sum_{k=1}^{d} \gamma_k = \Lambda$, $\sum_{k=1}^{d} \rho_k^2 \beta_k = \sum_{k=1}^{d} \rho_k^2 L_k \alpha_k^{L_k}$ and

$$\bar{\Gamma} = \prod_{k=1}^{d} \Gamma_k = \prod_{k=1}^{d} 2^{L_k} \alpha_k^{L_k^2 W_k} \left( \frac{2ep_k U_k}{W_k} \right)^{L_k W_k}$$

$$= 2^{\widetilde{L}} \prod_{k=1}^{d} \alpha_k^{L_k^2 W_k} \prod_{k=1}^{d} \left( \frac{2ep_k U_k}{W_k} \right)^{L_k W_k}$$

$$\leq 2^{\widetilde{L}} \left( \prod_{k=1}^{d} \alpha_k^{L_k^2 W_k} \right) \left( \frac{\sum_{k=1}^{d} 2eL_k W_k p_k U_k / W_k}{\sum_{k=1}^{d} L_k W_k} \right)^{\sum_{k=1}^{d} L_k W_k}$$

$$= 2^{\widetilde{L}} \left( \prod_{k=1}^{d} \alpha_k^{L_k^2 W_k} \right) \left( \frac{\sum_{k=1}^{d} 2ep_k L_k U_k}{\Lambda} \right)^{\Lambda},$$

where the inequality follows from the second inequality in Lemma A.1. Then, we have that $\gamma' = \widetilde{\gamma} + W = \Lambda + W$ and

$$\log(\Gamma') \leq \log \left( 2^{d} \bar{\rho}^{(\Lambda+W)(M+1)} 2^{\widetilde{L}} \left( \prod_{k=1}^{d} \alpha_k^{L_k^2 W_k} \right) \left( \frac{\sum_{k=1}^{d} 2ep_k L_k U_k}{\Lambda} \right)^{\Lambda} \left( \frac{e \sum_{k=1}^{d} \rho_k^2 \beta_k}{W} \right)^{W} \right)$$

$$= (d + \widetilde{L}) \log 2 + (\Lambda + W)(M+1) \log \bar{\rho} + \sum_{k=1}^{d} L_k^2 W_k \log \alpha_k + \Lambda \log \left( \sum_{k=1}^{d} 2ep_k L_k U_k \right)$$

$$- \Lambda \log \Lambda + W \log \left( e \sum_{k=1}^{d} \rho_k^2 L_k \alpha_k^{L_k} / W \right).$$

Therefore, we can apply Lemma 3.2 to obtain:

$\text{Pdim}\left( \mathcal{V}^{\text{MLP}} \right)$

$\leq 4 \left( (\Lambda + W) \log(2(\Lambda + W) + 1) + \log(2\Gamma') \right)$

$\leq 4 \left( \Lambda \log \left( \frac{2(\Lambda + W) + 1}{\Lambda} \right) + W \log \left( \frac{2(\Lambda + W) + 1}{W} \right) + (d + \widetilde{L} + 1) \log 2 \right.$

$\left. + (\Lambda + W)(M+1) \log \bar{\rho} + \sum_{k=1}^{d} L_k^2 W_k \log \alpha_k + \Lambda \log \left( \sum_{k=1}^{d} 2ep_k L_k U_k \right) + 2W \log \left( e\bar{\rho}\widetilde{L} \right) + W \log \left( \sum_{k=1}^{d} \alpha_k^{L_k} \right) \right)$

$\leq 4 \left( \Lambda \log 5 + W \log(3 + 2\widetilde{L}) + (d + \widetilde{L} + 1) \log 2 + 2\Lambda(M+1) \log \bar{\rho} + \Lambda \log \left( \sum_{k=1}^{d} 2ep_k L_k U_k \right) \right.$

$\left. + 2W \log(e\bar{\rho}) + 2W \log(\widetilde{L}) + W \log \left( \sum_{k=1}^{d} \alpha_k^{L_k} \right) + \sum_{k=1}^{d} L_k^2 W_k \log \alpha_k \right)$

$= \mathcal{O} \left( \Lambda + W \log \widetilde{L} + \Lambda M \log \bar{\rho} + \Lambda \log \left( \sum_{k=1}^{d} p_k L_k U_k \right) + W \log \bar{\rho} + W \log \left( \sum_{k=1}^{d} \alpha_k^{L_k} \right) + \sum_{k=1}^{d} L_k^2 W_k \log \alpha_k \right)$

$= \mathcal{O} \left( \Lambda M \log \bar{\rho} + \Lambda \log \left( \sum_{k=1}^{d} p_k U_k^2 \right) + W \log \left( \sum_{k=1}^{d} \alpha_k^{L_k} \right) + \sum_{k=1}^{d} L_k^2 W_k \log \alpha_k \right)$

$= \mathcal{O} \left( \left( \sum_{k=1}^{d} L_k W_k \right) \left( M \sum_{k=1}^{d} \log \rho_k + \log \left( \sum_{k=1}^{d} p_k U_k \right) \right) + W \log \left( \sum_{k=1}^{d} \alpha_k^{L_k} \right) + \sum_{k=1}^{d} L_k^2 W_k \log \alpha_k \right)$

For ReLU MLPs, we have $\alpha_k = 1$ and $p_k = 2$ for all $k \in [d]$. Thus, we can simplify the last term to obtain:

$$\text{Pdim}\left( \mathcal{V}^{\text{MLP}} \right) = \mathcal{O} \left( \left( \sum_{k=1}^{d} L_k W_k \right) \left( M \sum_{k=1}^{d} \log \rho_k + \log \left( \sum_{k=1}^{d} U_k \right) \right) \right)$$

$\square$

# C  Data-dependent sample complexity

## C.1  Proof of Proposition 3.9

**Lemma C.1** (Massart's Lemma). *Let $X = \{\mathbf{x}^1, \ldots, \mathbf{x}^r\} \subseteq \mathbb{R}^N$ be a finite set of vectors. Then we have*

$$\mathbb{E}_{\boldsymbol{\sigma} \sim \{-1,1\}^N} \left[ \max_{j \in [r]} \frac{1}{N} \langle \boldsymbol{\sigma}, \mathbf{x}^j \rangle \right] \leq \max_{j \in [r]} \left\| \mathbf{x}^j - \frac{1}{r} \sum_{i=1}^{r} \mathbf{x}^i \right\|_2 \frac{\sqrt{2 \log r}}{N}.$$

*Proof of Proposition 3.9.* We aim to bound the empirical Rademacher complexity $\widehat{\mathcal{R}}_{S_N}(\mathcal{V})$ for a sample $S_N = \{I_1, \ldots, I_N\} \subseteq \mathcal{I}$ with $N \geq \sum_{k=1}^{d}(\gamma_k + W_k)$:

$$\widehat{\mathcal{R}}_{S_N}(\mathcal{V}) = \mathbb{E}_{\boldsymbol{\sigma} \sim \{-1,1\}^N} \left[ \sup_{\mathbf{w} \in \mathcal{W}} \frac{1}{N} \sum_{i=1}^{N} \sigma_i V(I_i, \mathbf{w}) \right].$$

Let $X = \{(V(I_1, \mathbf{w}), \ldots, V(I_N, \mathbf{w})) \mid \mathbf{w} \in \mathcal{W}\}$ be the set of possible output vectors over the sample $S_N$. Lemma B.2 implies that the parameter space $\mathcal{W}$ can be partitioned into at most $r(S_N)$ regions, such that within each region, the vector $(V(I_1, \mathbf{w}), \ldots, V(I_N, \mathbf{w}))$ is constant. Therefore, the set $X$ is finite, containing at most $r(S_N)$ distinct vectors, say $X = \{\mathbf{x}^1, \ldots, \mathbf{x}^{r(S_N)}\}$. The supremum over $\mathbf{w} \in \mathcal{W}$ can thus be replaced by a maximum over the finite set $X$:

$$\widehat{\mathcal{R}}_{S_N}(\mathcal{V}) = \mathbb{E}_{\boldsymbol{\sigma} \sim \{-1,1\}^N} \left[ \max_{j \in [r(S_N)]} \frac{1}{N} \langle \boldsymbol{\sigma}, \mathbf{x}^j \rangle \right].$$

Applying Massart's Lemma (Lemma C.1), we get

$$\widehat{\mathcal{R}}_{S_N}(\mathcal{V}) \leq \max_{j \in [r(S_N)]} \left\| \mathbf{x}^j - \frac{1}{r(S_N)} \sum_{i=1}^{r(S_N)} \mathbf{x}^i \right\|_2 \frac{\sqrt{2 \log (r(S_N))}}{N}$$

$$\leq H\sqrt{N} \frac{\sqrt{2 \log (r(S_N))}}{N}$$

$$\leq H \sqrt{\frac{2}{N} \log \left( 2^d \bar{\Gamma} \left( \prod_{k=1}^{d} \widetilde{Q}_{M,k}^{\gamma_k} \right) \prod_{k=1}^{d} \left( \frac{e \widetilde{Q}_{M,k} \rho_k \beta_k}{W_k} \right)^{W_k} \right)}$$

$$= H \sqrt{\frac{2}{N} \left( d \log 2 + \log \bar{\Gamma} + \sum_{k=1}^{d} \gamma_k \log \widetilde{Q}_{M,k} + \sum_{k=1}^{d} W_k \log \left( \frac{e \widetilde{Q}_{M,k} \rho_k \beta_k}{W_k} \right) \right)}$$

$$= H \sqrt{\frac{2}{N} \left( d \log 2 + \log \bar{\Gamma} + \sum_{k=1}^{d} (\gamma_k + W_k) \log \widetilde{Q}_{M,k} + \sum_{k=1}^{d} W_k \log \left( \frac{e \rho_k \beta_k}{W_k} \right) \right)}$$

$$\leq H \sqrt{\frac{2}{N} \left( d \log 2 + \log \bar{\Gamma} + \sum_{k=1}^{d} (\gamma_k + W_k) \log \left( \sum_{k=1}^{d} \widetilde{Q}_{M,k} \right) + W \log \left( \frac{e \sum_{k=1}^{d} \rho_k \beta_k}{W} \right) \right)}$$

$$\leq H \sqrt{\frac{2}{N} \left( d + \sum_{k=1}^{d} \log \Gamma_k + (\widetilde{\gamma} + W) \log \left( \sum_{k=1}^{d} \sum_{i=1}^{N} Q_{M,k}(I_i) \right) + W \log \left( \frac{e \sum_{k=1}^{d} \rho_k \beta_k}{W} \right) \right)}.$$

The second inequality follows because $V(I_i, \mathbf{w}) \in [0, H]$ implies that each coordinate of $\mathbf{x}^j$ and $\frac{1}{r} \sum_{i=1}^{r} \mathbf{x}^i$ lies in $[0, H]$. Thus, each coordinate of the difference vector $\mathbf{x}^j - \frac{1}{r} \sum_{i=1}^{r} \mathbf{x}^i$ is bounded in absolute value by $H$. Since this vector is $N$-dimensional, its $L_2$ norm is therefore upper bounded by $H\sqrt{N}$. The third inequality is due to (4). $\qquad\square$

## C.2  Uniform convergence bounds from pseudo-dimension and Rademacher complexity

As discussed in Section 3.2, both pseudo-dimension and empirical Rademacher complexity yield uniform convergence guarantees, as stated in (1) and (2), respectively. The pseudo-dimension provides a data-independent measure of function class complexity. Its calculation (as in Theorem 3.3, leading to Propositions 3.4 and 3.8) ultimately incorporates worst-case estimates for the number of distinct state-action pairs, related to $\rho_k \bar{\rho}^M$ per instance (derived from Lemma B.1). In contrast, the empirical Rademacher complexity bound (Proposition 3.9) directly uses the empirically observed sum $\sum_{k=1}^d Q_{M,k}(I)$ for any instance $I \sim \mathcal{D}$. Since this sum can be substantially smaller than the worst-case estimate (i.e., $\rho_k \bar{\rho}^M$) in many settings, Rademacher-based bounds can be much tighter in practice. To facilitate a concrete comparison of the uniform convergence guarantees obtained by these two measures, we now consider the specific case where ReLU MLPs, characterized by $L_k, W_k, U_k$ for $k \in [d]$, serve as the scoring functions. Given $S_N = \{I_1, \ldots, I_N\} \overset{\text{i.i.d}}{\sim} \mathcal{D}^N$, substituting the pseudo-dimension bound for ReLU MLP policies from Proposition 3.8 into (1), we have:

$$
\sup_{\mathbf{w} \in \mathcal{W}} \left| \frac{1}{N} \sum_{i=1}^N V(I_i, \mathbf{w}) - \mathbb{E}_{I \sim D}[V(I_i, \mathbf{w})] \right|
$$

$$
= \mathcal{O}\left( H \sqrt{\frac{\mathrm{Pdim}(\mathcal{V}) + \log(1/\delta)}{N}} \right)
$$

$$
= \mathcal{O}\left( H \sqrt{\frac{\left( \sum_{k=1}^d L_k W_k \right) \left( M \sum_{k=1}^d \log \rho_k + \log \left( \sum_{k=1}^d U_k \right) \right)}{N}} + H \sqrt{\frac{\log(1/\delta)}{N}} \right).
$$

To derive the empirical Rademacher complexity bound for policies employing $d$ ReLU MLP scoring functions, we apply the results of Lemma 3.7 by taking $\alpha_k = 1$ and $p_k = 2$ (characteristic of ReLU). Let $\Lambda = \sum_{k=1}^d L_k W_k$ and $\widetilde{L} = \sum_{k=1}^d L_k$, and for this MLP case $\widetilde{\gamma} = \Lambda$ based on Lemma 3.7. Substituting these ReLU-specific structural parameters $(\Gamma_k, \gamma_k, \beta_k)$ into the general empirical Rademacher complexity bound from Proposition 3.9 yields:

$$
\widehat{\mathcal{R}}_{S_N}(\mathcal{V})
$$

$$
\leq H \sqrt{\frac{2}{N} \left( d + \widetilde{L} + \Lambda \log \left( \frac{\sum_{k=1}^d L_k U_k}{\Lambda} \right) + (\Lambda + W) \log \left( \sum_{k=1}^d \sum_{i=1}^N Q_{M,k}(I_i) \right) + W \log \left( \frac{e \sum_{k=1}^d \rho_k L_k}{W} \right) \right)}
$$

$$
\leq H \sqrt{\frac{2}{N} \left( d + \widetilde{L} + (\Lambda + W) \log \left( \sum_{k=1}^d \sum_{i=1}^N Q_{M,k}(I_i) \right) + W \log \left( e \sum_{k=1}^d \rho_k \right) \right)}
$$

$$
= \mathcal{O}\left( H \sqrt{\frac{1}{N} \left( \Lambda \log \left( \sum_{k=1}^d \sum_{i=1}^N Q_{M,k}(I_i) \right) + W \log \left( \sum_{k=1}^d \rho_k \right) \right)} \right)
$$

Therefore, substituting this into (2), we obtain the following bound with probability at least $1 - \delta$:

$$
\sup_{\mathbf{w} \in \mathcal{W}} \left| \frac{1}{N} \sum_{i=1}^N V(I_i, \mathbf{w}) - \mathbb{E}_{I \sim D}[V(I_i, \mathbf{w})] \right|
$$

$$
= \mathcal{O}\left( \widehat{\mathcal{R}}_{S_N}(\mathcal{V}) + H \sqrt{\frac{\log(1/\delta)}{N}} \right)
$$

$$
= \mathcal{O}\left( H \sqrt{\frac{\left( \sum_{k=1}^d L_k W_k \right) \log \left( \sum_{k=1}^d \sum_{i=1}^N Q_{M,k}(I_i) \right) + W \log \left( \sum_{k=1}^d \rho_k \right)}{N}} + H \sqrt{\frac{\log(1/\delta)}{N}} \right).
$$

$$
\tag{6}
$$

The preceding uniform convergence bounds for ReLU MLP policies highlight the key distinction when comparing their dominant terms. The pseudo-dimension based bound (1), via Proposition 3.8, is influenced by factors scaling with $M \sum_{k=1}^{d} \log \rho_k$. This term reflects a dependency on the maximum rounds $M$ and the maximum action space sizes, characteristic of worst-case, data-independent analysis. In contrast, the Rademacher-based bound (6) has the corresponding leading term $\log \left( \sum_{k=1}^{d} \sum_{i=1}^{N} Q_{M,k}(I_i) \right)$. This shows a dependency on the logarithm of the empirically observed total number of distinct state-action pairs. As this empirical sum can be substantially smaller than the quantities implied by the worst-case parameters $M$ and $\bar{\rho}$ in many practical scenarios, the Rademacher-based bound may offer a tighter guarantee.

As an example, consider a set of binary integer programming problems, each with variables $\mathbf{x}_j \in \{0, 1\}$ for $j \in [n]$. Assume that for these problems, the feasible regions of their respective LP relaxations all have an empty intersection with the boundary of the $[0, 1]^n$ hypercube. Suppose that only the variable branching policy is learned (i.e., $d = 1$, with $\rho_1 = n$), while other decisions are deterministic. In such a scenario, any variable branching operation can lead to the termination of Algorithm 2 in the context of B&C by proving infeasibility for the resulting subproblems. Then, the total number of distinct branching decision contexts (state-action pairs) for an instance $I_i$ is on the order of $O(n)$. However, the pseudo-dimension based bound would still have the term $M \log n$. This term could be much larger than a term like $\log(Nn)$, which would be the leading term of the bound derived from empirical Rademacher complexity in this case (since $\sum_{i=1}^{N} Q_{M,1}(I_i) = O(Nn)$). Consequently, the empirical Rademacher complexity's direct dependence on $\sum_{i=1}^{N} Q_{M,1}(I_i)$ would yield a tighter sample complexity guarantee.

In a similar vein, if a tighter universal bound on $Q_{M,k}(I)$ for all instances $I$ can be established—based on specific properties or structural characteristics of problems drawn from the distribution under study—that is better the worst possible $\rho_k \bar{\rho}^M$ estimate (which was used in Lemma B.1), then the bounds in Theorem 3.3 can be improved. Consequently, this would also lead to more refined pseudo-dimension bounds in Propositions 3.4 and 3.8. A simple bound is $|\mathcal{S}|$, the total number of states in the state space $\mathcal{S}$. In certain settings, $|\mathcal{S}|$ may be much smaller than $\rho_k \bar{\rho}^M$ because the latter counts different paths taken in the decision process which can be exponentially larger than the total number of states. Another situation is when the termination states/conditions in the decision process imply a smaller value for $Q_{M,k}(I)$ for all $I$, compared to $\rho_k \bar{\rho}^M$ or $|\mathcal{S}|$.

While bounds based on the empirical Rademacher complexity, $\widehat{\mathcal{R}}_{S_N}(\mathcal{V})$, are valuable as they adapt to the specific set of instances $S_N$ (e.g., via terms involving instance-specific quantities $Q_{M,k}(I_i)$ for $I_i \in S_N$), this instance-specificity means their values would be hard to estimate for a different set of instances $S'_N \sim \mathcal{D}^N$ (e.g., when considering a new test set). To obtain a bound that is characteristic of the distribution $\mathcal{D}$ (for a given sample size $N$), we use the expected Rademacher complexity, $\mathcal{R}_N(\mathcal{V}) = \mathbb{E}_{S_N \sim \mathcal{D}^N}[\widehat{\mathcal{R}}_{S_N}(\mathcal{V})]$. This quantity reflects the average-case complexity over all possible samples, depending on distribution-dependent quantities like $\mu_{M,k} = \mathbb{E}_{I \sim \mathcal{D}}[Q_{M,k}(I)]$, rather than instance-specific values from a particular sample. Formally, this derivation proceeds as follows:

$$\mathcal{R}_N(\mathcal{V}) = \mathbb{E}_{S_N \sim \mathcal{D}^N} \left[ \widehat{\mathcal{R}}_{S_N}(\mathcal{V}) \right]$$

$$\leq \mathbb{E}_{S_N \sim \mathcal{D}^N} \left[ H \sqrt{\frac{2}{N} \left( d + \widetilde{L} + (\Lambda + W) \log \left( \sum_{k=1}^{d} \sum_{i=1}^{N} Q_{M,k}(I_i) \right) + W \log \left( e \sum_{k=1}^{d} \rho_k \right) \right)} \right]$$

$$\leq H \sqrt{\frac{2}{N} \left( d + \widetilde{L} + (\Lambda + W) \mathbb{E}_{S_N \sim \mathcal{D}^N} \left[ \log \left( \sum_{k=1}^{d} \sum_{i=1}^{N} Q_{M,k}(I_i) \right) \right] + W \log \left( e \sum_{k=1}^{d} \rho_k \right) \right)}$$

$$\leq H \sqrt{\frac{2}{N} \left( d + \widetilde{L} + (\Lambda + W) \log \left( \mathbb{E}_{S_N \sim \mathcal{D}^N} \left[ \sum_{k=1}^{d} \sum_{i=1}^{N} Q_{M,k}(I_i) \right] \right) + W \log \left( e \sum_{k=1}^{d} \rho_k \right) \right)}$$

$$= H \sqrt{\frac{2}{N} \left( d + \widetilde{L} + (\Lambda + W) \log \left( N \sum_{k=1}^{d} \mu_{M,k} \right) + W \log \left( e \sum_{k=1}^{d} \rho_k \right) \right)},$$

where the second and third inequalities utilized Jensen's inequality. Then, standard uniform convergence results (e.g., [Mohri et al., 2018, Theorem 3.3]) imply that with probability at least $1 - \delta$:

$$\sup_{\mathbf{w} \in \mathcal{W}} \left| \frac{1}{N} \sum_{i=1}^{N} V(I_i, \mathbf{w}) - \mathbb{E}_{I \sim \mathcal{D}}[V(I, \mathbf{w})] \right|$$

$$= \mathcal{O} \left( \mathcal{R}_N(\mathcal{V}) + H \sqrt{\frac{\log(1/\delta)}{N}} \right)$$

$$= \mathcal{O} \left( H \sqrt{\frac{\left( \sum_{k=1}^{d} L_k W_k \right) \log \left( N \sum_{k=1}^{d} \mu_{M,k} \right) + W \log \left( \sum_{k=1}^{d} \rho_k \right)}{N}} + H \sqrt{\frac{\log(1/\delta)}{N}} \right).$$

