# OpenReview forum: "Generalization Guarantees for Learning Score-Based Branch-and-Cut Policies in Integer Programming"
_NeurIPS.cc/2025/Conference — NeurIPS 2025 poster_

### Official Review · Reviewer_twK1 · 2025-06-17

**Clarity:** 3
**Significance:** 3
**Originality:** 2
**Rating:** 5
**Confidence:** 5

**Summary:**

This paper contributes to a recent line of work on provable sample-complexity guarantees for learning to tune branch-and-cut. Prior work by Balcan et al. has focused on learning linear weights to combine predetermined heuristic scores such as efficacy, parallelism, etc. in order to choose between branch-and-cut actions (such as what cuts to add, what variable to branch on, what node to explore next, and so on). The present work builds and extends upon that line of work by learning scores that are piecewise polynomial functions of the parameter weights. This is better reflective of current practice that has used neural networks to map instances (the state of a branch-and-cut solve of an integer program) to desired actions (the next cut to add, etc.) directly. The bounds are obtained by analyzing the pseudodimension of the class of induced cost functions, and the authors do this for a general abstraction of a sequential decision process that generalizes branch-and-cut.

**Questions:**

In [1] the authors also consider a general abstraction of tree search that generalizes branch-and-cut, and their main sample complexity results are for that general model of tree search. How does that model compare to the model of a sequential decision process that you consider in the present paper? (I see that the authors in [1] also only consider linear weightings of scores, but a comparison of the two models would nonetheless be useful.) Also, how does the model compare to the formal model of dynamic programming?

Definition 3.1 to me seems like a special case of the general notion of piecewise decomposability in [2]. Is that correct? If so there should probably be some mention of this.

Very minor nitpick on notation: I think using k = 1, 2, 3 to denote node selection, cuts, and branching is awkward and you should just write something like A^s_{node}, A^s_{cut}, A^s_{branch} or something like that.


[1] Balcan, Prasad, Sandholm, Vitercik. Improved sample complexity bounds for branch-and-cut. CP 2022.

[2] Balcan, Deblasio, Dick, Kingsford, Sandholm, Vitercik. How Much Data Is Sufficient to Learn High-Performing Algorithms? Journal of the ACM.

**Ethical Concerns:**

["NO or VERY MINOR ethics concerns only"]

**Final Justification:**

The authors provided a detailed rebuttal. I am in favor of accepting the paper.

**Limitations:**

Yes

**Paper Formatting Concerns:**

None.

**Quality:**

3

**Strengths And Weaknesses:**

This is a nice theory paper that contributes a missing piece to the recent generalization theory for learning cut selection policies. It provides a theoretical backing for various experimental papers. The proofs mostly seem to rely on by-now standard techniques, but that is ok. Some experiments showing the impact of parameter tuning and/or empirically testing the sample complexity would be nice, but not required for a theory paper.

One weakness is that the work could be seen as largely incremental to prior work, with no fundamentally new techniques introduced. But I also think that is OK.

See the question section for a few specific comments.

---

> ### Author Rebuttal · Authors · 2025-07-30
>
> Thank you very much for the thoughtful and encouraging review.
>
> > Some experiments showing the impact of parameter tuning and/or empirically testing the sample complexity would be nice, but not required for a theory paper.
>
> We did not include experiments in the original submission as our primary motivation was to provide the theoretical foundations for the successful empirical works surveyed in Section 2.1. However, we agree with the reviewers that a computational study strengthens the paper. We have thus conducted a new proof-of-concept experiment to provide an empirical view of our theoretical bounds. The results verify the predicted $\mathcal{O}(1/\sqrt{N})$ convergence rate ($R^2=0.944$) and show that practical generalization errors are substantially smaller than the worst-case bounds. For the full details of this experiment, we respectfully refer the reviewer to our response to Reviewer U6oB.
>
> >In [1] the authors also consider a general abstraction of tree search that generalizes Branch-and-Cut (B&C), and their main sample complexity results are for that general model of tree search. How does that model compare to the model of a sequential decision process that you consider in the present paper? (I see that the authors in [1] also only consider linear weightings of scores, but a comparison of the two models would nonetheless be useful.)
>
> The framework in [1] and our sequential decision process differ in two fundamental aspects:
> * The model in [1] analyzes policies derived from a convex combination of two pre-defined scores, parameterized by a single scalar. Our framework provides guarantees for a much broader class of policies where the scoring function itself is a general piecewise polynomial function of a high-dimensional parameter vector $\mathbf{w}$—a framework that, as we detail in our response to Reviewer SAVB, can be further extended to the even more general class of piecewise Pfaffian functions.
>
> * The analysis in [1] crucially relies on the scoring functions and action sets to have a certain "path-wise" property to achieve its strong, depth-dependent bounds. Our framework is more general as it is state-based and does not require this structural assumption. This is a critical distinction, as many sophisticated heuristics used in empirical work are not strictly path-wise; for example, scoring rules may incorporate global tree features (e.g., the current primal-dual gap or the total number of open nodes), which our state-based model can naturally accommodate.
>
> > Also, how does the model compare to the formal model of dynamic programming?
>
>
> Our model is exactly the same as the formal model of decision making studied within (deterministic) dynamic programming (DP). Of course, for something like B&C, one would not want to use DP to decide on the optimal set of actions at every decision point as this would be prohibitively expensive from a computational standpoint. Instead we use the framework as a formal language to analyze data-driven learning paradigms used to make these decisions within B&C. Naturally, our general framework allows applications to data-driven (as opposed to DP based) approaches for sequential decision making beyond B&C.
>
> We thank the reviewer for these valuable suggestions. In the final version, we will add a detailed discussion situating our framework with respect to both the tree search model in [1] and the broader literature on Markov Decision Process (MDP) formulations for B&C, as well as dynamic programming.
>
> > Definition 3.1 to me seems like a special case of the general notion of piecewise decomposability in [2]. Is that correct? If so there should probably be some mention of this.
>
> This is an insightful question, and indeed there is a conceptual connection between our Definition 3.1 and the notion of "piecewise decomposability" in [2]. The $(\mathcal{F}, \mathcal{G}, k)$-structure from [2] is more general as it allows $\mathcal{F}$ to be any class with a finite dual pseudo-dimension, whereas our work focuses on piecewise polynomial function classes. However, this specialization is well-motivated, as many applications of the framework in [2] and subsequent works ultimately consider much simpler settings, such as piecewise constant or linear functions (e.g., [1, 2, 3]).
>
> The primary advantage of our $(\Gamma, \gamma, \beta)$-structure, compared to the one in [2], lies in how it characterizes the partition of the parameter space, which could be a more direct and natural setting for certain function classes.
>
> * First, describing the partition via $k$ functions drawn from a single boundary class $\mathcal{G}$ (as done in [2]) is not always ideal. For a ReLU Multi-Layer Perceptron (MLP), the parameter space is partitioned by an arrangement of polynomials with varying degrees (from $1$ to $L$, the depth of the MLP). A direct, layered analysis like in [4] can establish sharper bounds on the number of regions, compared to a potentially lossy analysis that fits a single boundary class $\mathcal{G}$ to describe this complex arrangement. Doing so could lose information by forcing all boundary polynomials to have the highest degree.
>
> * Second, for function classes such as piecewise polynomials and MLPs, applying the framework of [2] can result in another source of loss in the tightness of the bounds on pseudo-dimension. The framework in [2] takes as given the VC dimension of the boundary functions defining the pieces, and applies Sauer's Lemma to derive a bound on the number of expressible sign patterns (i.e., pieces or regions). If a tighter, direct bound on the number of pieces is available (as in our Lemma 3.7), using it usually gives a strictly better upper bound on the overall pseudo-dimension. This potential loss in tightness has been noted in several recent works, including our own Remark 3.5 where a concrete example is discussed, and in the discussions in [6, 7, 8, 9].
>
> In summary, while conceptually related to [2], our definition is tailored for settings with piecewise polynomials, leading to a more efficient characterization and tighter bounds for this important class of problems. We will add a discussion of this relationship in the final version.
>
>
> > Very minor nitpick on notation: I think using k = 1, 2, 3 to denote node selection, cuts, and branching is awkward and you should just write something like $A^s_{node}, A^s_{cut}, A^s_{branch}$ or something like that.
>
> This is a great suggestion for improving clarity. We agree that the proposed notation is much more intuitive, and we will adopt it in the final version of the paper.
>
> ---
>
> [1] Balcan, M.F., Prasad, S., Sandholm, T. and Vitercik, E., 2022, July. Improved Sample Complexity Bounds for Branch-And-Cut. In 28th International Conference on Principles and Practice of Constraint Programming.
>
> [2] Balcan, M.F., Deblasio, D., Dick, T., Kingsford, C., Sandholm, T. and Vitercik, E., 2024. How much data is sufficient to learn high-performing algorithms?. Journal of the ACM, 71(5), pp.1-58.
>
> [3] Balcan, M.F.F., Prasad, S., Sandholm, T. and Vitercik, E., 2022. Structural analysis of branch-and-cut and the learnability of gomory mixed integer cuts. Advances in Neural Information Processing Systems, 35, pp.33890-33903.
>
> [4] Bartlett, P.L., Harvey, N., Liaw, C. and Mehrabian, A., 2019. Nearly-tight VC-dimension and pseudodimension bounds for piecewise linear neural networks. Journal of Machine Learning Research, 20(63), pp.1-17.
>
> [5] Anthony, M. and Bartlett, P.L., 2009. Neural network learning: Theoretical foundations. cambridge university press.
>
> [6] Bartlett, P., Indyk, P. and Wagner, T., 2022, June. Generalization bounds for data-driven numerical linear algebra. In Conference on Learning Theory (pp. 2013-2040). PMLR.
>
> [7] Balcan, M.F., Nguyen, A.T. and Sharma, D., Algorithm Configuration for Structured Pfaffian Settings. Transactions on Machine Learning Research.
>
> [8] Cheng, H. and Basu, A., 2024. Learning cut generating functions for integer programming. Advances in Neural Information Processing Systems, 37, pp.61455-61480.
>
> [9] Cheng, H., Khalife, S., Fiedorowicz, B. and Basu, A., 2024. Sample complexity of algorithm selection using neural networks and its applications to branch-and-cut. Advances in Neural Information Processing Systems, 37, pp.25036-25060.

---

> > ### Comment · Reviewer_twK1 · 2025-08-03
> >
> > Thanks to the authors for their detailed response.

---

### Official Review · Reviewer_u1pW · 2025-06-23

**Clarity:** 3
**Significance:** 3
**Originality:** 3
**Rating:** 5
**Confidence:** 2

**Summary:**

This paper presents a rigorous theoretical framework to analyze the generalization performance of learned heuristic policies for Branch-and-Cut (B&C) algorithms in Mixed Integer Programming (MIP). It introduces sample complexity bounds for policies guided by piecewise polynomial scoring functions, a class that includes linear models and neural networks (e.g., MLPs with ReLU).

**Questions:**

(1) How sensitive are your generalization guarantees to the expert quality (e.g., strong branching vs. suboptimal heuristic)?
(2) How do the theoretical bounds scale in practice with real-world ReLU network sizes used in B&C solvers?

**Ethical Concerns:**

["NO or VERY MINOR ethics concerns only"]

**Final Justification:**

The author provided detailed explanations to my questions, and I will maintain my score for this paper.

**Limitations:**

(1) Role of oracle quality in training (e.g., strong branching) could influence generalization.

**Quality:**

3

**Strengths And Weaknesses:**

The strength of this paper:

1. extends previous PAC-style algorithm configuration theory, yielding new pseudo-dimension and Rademacher complexity bounds for both worst-case and data-dependent settings. It bridges the gap between empirical methods and theoretical guarantees.

The weakness of this paper:

1. Lacks guidance on how to translate theory into practice (e.g., how to bound pseudo-dimension for specific networks in practice). Providing case studies may facilitate the reader's better understanding of the claims in this paper.

---

> ### Author Rebuttal · Authors · 2025-07-30
>
> Thank you very much for the thoughtful review.
>
> **Weakness:**
>
> We would like to clarify that for the class of Multi-Layer Perceptron (MLP) policies with piecewise polynomial activations (e.g., ReLU), our paper already provides a direct translation of this nature. Specifically, Proposition 3.8 and 4.2 offer explicit pseudo-dimension bounds directly in terms of a network's architectural characteristics (number of parameters $W$, layers $L$, neurons $U$, etc.). More fine-tuned bounds can be given in terms of the specific sizes of the different layers, which we left out for space and clarity of the stated bounds. Our intent with these results was to provide precisely the kind of guidance the reviewer is asking for: a clear mapping from practical model design choices to their theoretical sample complexity implications.
>
> We fully agree with the reviewer that extending this to a broader range of architectures is an excellent point that would further strengthen the paper. As mentioned in our response to Reviewer SAVB, our framework can be generalized to cover models with "piecewise Pfaffian" structure. When we add the promised discussion section on this Pfaffian extension to the final version, we will be sure to include concrete case studies, outlining how to derive bounds for specific, empirically-used models involving Graph Neural Networks (GNNs) and/or attention mechanisms.
>
> **Questions:**
>
> 1. This question points to the important distinction between generalization error and approximation error. Our generalization guarantees provide an upper bound on the *uniform convergence error* for the entire function class parametrized by $\mathcal{W}:$ $\sup_{\mathbf{w} \in \mathcal{W}}\left|\frac{1}{N} \sum_{i=1}^N V\left(I_i, \mathbf{w}\right)-\mathbb{E}_{I \sim \mathcal{D}}[V(I, \mathbf{w})]\right|.$
>
>     This bound is determined by the complexity of the function class $\mathcal{V}$ as defined on lines 155-156 of the paper, which can be measured in a distribution-independent manner by its pseudo-dimension or in a distribution-dependent manner by its Rademacher complexity. Crucially, both measures of complexity depend on the policy architecture but are independent of the expert that generates the training labels. Our guarantee thus ensures that for any policy $\mathbf{w} \in \mathcal{W}$ (whether trained using an expert or some other means), its empirical performance on the sample is a reliable estimate of its true expected performance.
>
>     The expert's quality, however, is critical for a different aspect: the empirical performance on the sample $S$ of the specific policy $\widehat{\mathbf{w}}\_S$ selected  after the training process. Our theoretical guarantee works as follows:
>
>     * A high-quality expert will guide the learner to a policy $\widehat{\mathbf{w}}_S$ with low empirical cost. Our guarantee ensures that this empirically effective policy will also have a low true expected cost, i.e., generalize well to unseen instances.
>
>     * Conversely, a suboptimal expert will lead to a policy with high empirical cost. Our guarantee still holds, but simply confirms that this empirically poor policy will also perform poorly in expectation on new instances.
>
>     Thus, our framework provides guarantees for the *relative* quality of the final learned policy (on unseen instances) with respect to its empirical performance on the sample, while the *overall* performance of the final learned policy is naturally bounded by the quality of the expert being imitated.
>
> 2. The primary message of our theoretical bounds is to provide a "scaling law" that governs the relationship between sample complexity and network size for learning B\&C policies. As established in Propositions 3.8, 3.9, 4.1 and 4.2, the dominant term in our pseudo-dimension bound scales linearly with the number of network parameters ($W$) and layers ($L$).
>
>     This provides direct, actionable guidance: if a practitioner has a trained model that already achieves a satisfactory performance, but wishes to increase its expressive power with a larger architecture to further reduce the inductive bias, our results indicate that maintaining a similar generalization guarantee requires that the number of training instances increase by a commensurate factor. This theoretical principle, derived from our specific analysis of the sequential decision process, establishes a clear and practical trade off between model complexity and data requirements in this domain.
>
>     We also conducted a new proof-of-concept experiment to provide an empirical view of our theoretical bounds. The results verify the predicted $\mathcal{O}(1/\sqrt{N})$ convergence rate ($R^2=0.944$) and show that practical generalization errors are substantially smaller than the worst-case bounds. For the full details of this experiment, we respectfully refer the reviewer to our response to Reviewer U6oB.
>
>
> **Limitations:** This relates directly to our previous response concerning the distinction between generalization and approximation error. As detailed there, our guarantees are independent of the expert's quality, as they give bounds on the *relative* quality of the learned policy (on unseen instances) with respect to its empirical performance on the sample.

---

> > ### Comment · Reviewer_u1pW · 2025-08-04
> >
> > Thanks to the authors for their clarification.

---

### Official Review · Reviewer_U6oB · 2025-07-01

**Clarity:** 3
**Significance:** 2
**Originality:** 4
**Rating:** 5
**Confidence:** 2

**Summary:**

This paper develops a theoretical framework for analyzing the generalization guarantees of learned policies for Branch and Cut (B&C) when solving mixed-integer programs (MIPs). B&C is one of the most widely used techniques for solving MIPs. Existing solvers primarily rely on heuristics for node selection, cut selection, and branching variable selection. Recently, there has been growing interest in learning B&C policies. This paper contributes to this line of work by providing sample complexity bounds for learning such policies.



The authors view B&C as a sequential decision-making process involving three types of actions at each step: node selection, cut selection, and branching variable selection. The goal is to learn a parametric scoring function that guides these actions. To analyze the difficulty of learning such policies, the authors derive sample complexity upper bounds for both linear and multilayer perceptron (MLP) scoring functions.

The framework presented in Section 3 provides a general analysis of imitation learning, with learning B&C policies treated as a specific application of imitation learning. Finally Section 4 provides bound for B&C from the bounds developed in Section 3.

**Questions:**

- In the context of Branch-and-Cut, what exactly does the penalty function measure? Could you clarify its practical motivation—for example, how it reflects solver performance or computational cost?

- How does the complexity analysis compare to the analysis provided in Theorem 5.1 of Cheng, Basu (NeurIPS 2024)?

- How might your theoretical findings impact  or strengthen existing learning-based B&C methods such as those proposed by Tang et al. (ICML,2024), Khalil el al (AAAI, 2016), Cheng, Basu (NeurIPS 2024)?

**Ethical Concerns:**

["NO or VERY MINOR ethics concerns only"]

**Final Justification:**

I recommend accepting this paper. Prior to the rebuttal, I was already inclined to accept. During the rebuttal, the authors provided thorough responses to all of my questions. They also conducted and reported an additional computational study, which further strengthens the contribution. Hence, I will be maintaining my rating in favor of acceptance.

**Limitations:**

YES

**Quality:**

3

**Strengths And Weaknesses:**

I am not entirely familiar with the research topic. I am aware of some recent works which aim to learn policies for solving MIP problems. This paper, on the other hand, makes a strong theoretical contribution and provides a generic framework for sample complexity bounds for learning B&C policies

While the theoretical framework is well-developed, the paper could benefit from a computational study or empirical experiments.

---

> ### Author Rebuttal · Authors · 2025-07-30
>
> Thank you very much for the thoughtful review.
>
>
> **Experiments:**
>
> We did not include experiments in the original submission as our primary motivation was to provide the theoretical foundations for the successful empirical works surveyed in Section 2.1. However, we agree with the reviewers that a computational study strengthens the paper. We have thus conducted a new experiment to empirically investigate the convergence behavior of the generalization error.
>
> We first trained a fixed cut-selection policy. This policy is implemented as a Multi-Layer Perceptron (MLP) with two input neurons (for cut efficacy and parallelism), two hidden layers with five neurons each using ReLU activations, and a single output neuron with a clipped ReLU activation to produce a score in $[0,1]$. The network, parameterized by $\mathbf{w}$, was trained to imitate an expert signal based on the Linear Programming (LP) gap closed by a cut. We then evaluated this fixed policy on the packing distribution from [3], with 15 constraints and 30 variables. For each instance $I$, the performance metric $V(I, \mathbf{w})$ is the Branch-and-Cut (B&C) tree size resulting from a procedure where our policy adds 3 cuts per round for 5 consecutive rounds at the root node.
>
> The quantity we measured is $f(N) = \left|\frac{1}{N}\sum_{i \in S^{\text{train}}\_{N}} V(I_i, \mathbf{w}) - \frac{1}{N}\sum_{j \in S^{\\text{test}}\_{N}} V(I_j, \mathbf{w})\right|.$ For each sample size $N$ (from 1 to 50), the sets $S^{\text{train}}\_{N}$ and $S^{\text{test}}_{N}$ consist of $N$ instances drawn independently and uniformly at random from their respective larger pools of training and unseen test instances. To ensure statistical stability, we repeated this sampling process $10$ times and report the average $f(N)$. The motivation for using this quantity is theoretically grounded. Our theory (e.g., Eq. (1)) bounds the deviation of both the training and test (of size $N$) averages from the true expectation, $\mathbb{E}[V(I, \mathbf{w})]$, by a term that scales as $\mathcal{O}(1/\sqrt{N})$. It then follows from the triangle inequality that their difference, $f(N)$, is also bounded by a term of the same order. This makes $f(N)$ a direct empirical quantity for investigating the predicted convergence rate.
>
> It is important to clarify that the goal of this experiment is a proof-of-concept for the rate of convergence, not to numerically verify the specific constants in the theoretical upper bound. To this end, we fitted the empirical data to a curve with the theoretically-motivated functional form $g(N) = a \sqrt{1/N}$, yielding the parameter $a=346.86$. The results indicate a strong correspondence between our theory and this empirical observation, as the fitted curve achieved a high coefficient of determination of $R^2 = 0.944$. Due to formatting limitations, we cannot include the plot, but the table below presents a comparison for several representative sample sizes, showing the general close match between the empirical error $f(N)$ and the fitted curve $g(N)$.
>
> | $N$ | $f(N)$ (avg) | $g(N)$ (fitted) | Relative Error |
> |---:|---:|---:|---:|
> | 1 | 337.90 | 346.86 | 2.65% |
> | 2 | 289.75 | 245.27 | 15.35% |
> | 3 | 176.77 | 200.26 | 13.29% |
> | 4 | 152.00 | 173.43 | 14.10% |
> | 5 | 126.18 | 155.12 | 22.94% |
> | 10 | 100.09 | 109.69 | 9.59% |
> | 20 | 81.19 | 77.56 | 4.47% |
> | 30 | 65.17 | 63.33 | 2.82% |
> | 40 | 56.09 | 54.84 | 2.21% |
> | 50 | 38.26 | 49.05 | 28.20% |
>
> This experiment provides strong empirical evidence that the generalization error of a learned B&C policy for this distribution converges at the rate predicted by our theoretical analysis. We believe this result strengthens the connection between our theory and solver practice, and we will include and extend this empirical study in the final version of the paper.
>
>
> **Questions:**
>
> 1. The penalty function $P_k(s, a, i)$ in our framework is a general mechanism designed to quantify the immediate local cost associated with taking action $a$ (of type $k$) in state $s$ at round $i$. Its practical motivation is to allow for the modeling of various concrete measures of solver performance or computational cost. In the specific application to B&C presented in Section 4, we instantiate the penalty function to measure the size of the B&C tree, a standard measure of solver efficiency. For example, by setting the penalty to 1 for a cut action and 2 for a branch action, the total accumulated penalty $V(I, \mathbf{w})$ directly corresponds to the total number of nodes explored.
>
>     However, a key strength of our framework is that our core theoretical results (e.g., Theorem 3.3) impose no restrictions on the functional form of $P_k$. As long as it is a deterministic function mapping the $(s,a,i)$ triplet to a real-valued cost, our analysis holds. This allows one to define $P_k$ to model other important computational costs, such as the CPU time for solving an LP relaxation or for performing cut separation, under the standard assumption that an identical action in an identical state incurs an identical cost.
> 2. There are several key differences between the learning paradigm considered in our paper and that of Cheng and Basu [1]:
>     * Our work analyzes the paradigm of learning a *scoring function*, which assigns a score to each available action, with the highest-scoring action being selected. This aligns with the prevalent practice in modern solvers and much of the empirical literature. In contrast, [1] considers learning a neural network that directly maps a problem instance to a specific action (e.g., a cutting plane parameter).
>
>     * The analysis in [1] is focused on the important special case of adding cutting planes only at the root node of the B&C tree. Our framework is more general, providing guarantees for a sequential process where decisions (node selection, cut selection, and branching) are made dynamically throughout the entire B&C tree, not just at the root.
>
>     * Cheng and Basu [1] analyzes a setting with a continuous, infinite action space, as they consider an infinite family of cutting planes. Our paper, in line with common solver implementations like SCIP and most empirical studies, assumes that the set of available actions at any given state is finite.
>
> 3. This work aims to provide the rigorous theoretical foundations that underpin the empirical success of many learning-based B&C methods, such as those surveyed in Section 2.1 of the paper. For foundational empirical studies like [2], our framework provides what we believe to be the first formal generalization guarantees for learning the types of score-based policies they investigate. While we are uncertain of the specific Tang et al. (ICML, 2024) paper being referred to, we believe our general framework is a valuable tool for analyzing the generalization guarantees of a wide range of such data-driven policies.
>
>     Another insight from our analysis that we think could be valuable for practitioners is the following. Our work provides a "scaling law" for neural network design in this setting: the sample complexity scales linearly with the number of network parameters ($W$) and layers ($L$), as shown in Propositions 3.8, 3.9, 4.1 and 4.2. This provides direct, actionable guidance: if a practitioner has a trained model that already achieves a satisfactory performance, but wishes to increase its expressive power with a larger architecture to further reduce the inductive bias, our results indicate that maintaining a similar generalization guarantee requires that the number of training instances increase by a commensurate factor. This theoretical principle, derived from our specific analysis of the sequential decision process, establishes a clear and practical trade off between model complexity and data requirements in this domain.
>
> ---
>
> [1] Cheng, H. and Basu, A., 2024. Learning cut generating functions for integer programming. Advances in Neural Information Processing Systems, 37, pp.61455-61480.
>
> [2] Khalil, E., Le Bodic, P., Song, L., Nemhauser, G. and Dilkina, B., 2016, February. Learning to branch in mixed integer programming. In Proceedings of the AAAI conference on artificial intelligence (Vol. 30, No. 1).
>
> [3] Tang, Y., Agrawal, S. and Faenza, Y., 2020, November. Reinforcement learning for integer programming: Learning to cut. In International conference on machine learning (pp. 9367-9376). PMLR.

---

> > ### Comment · Reviewer_U6oB · 2025-08-04
> > **Technically solid paper**
> >
> > I thank the authors for the detailed response. I also read the the points of other reviewers and am convinced that the paper is a high-impact paper with a strong theoretical contribution.

---

### Official Review · Reviewer_SAVB · 2025-07-02

**Clarity:** 3
**Significance:** 2
**Originality:** 2
**Rating:** 5
**Confidence:** 2

**Summary:**

This paper provides a theoretical analysis of the sample complexity for learning heuristic policies within the Branch-and-Cut (B&C) framework for mixed-integer programming. The authors introduce a general model for sequential decision-making that abstracts the B&C process. Their central contribution is to show that if the scoring functions guiding decisions (e.g., node, cut, and variable selection) have a "piecewise polynomial" structure with respect to their learnable parameters, then the overall performance metric (e.g., B&C tree size) is a "piecewise constant" function of those same parameters. This structural insight allows them to derive novel generalization bounds, via pseudo-dimension and Rademacher complexity, for policies parameterized not only by traditional linear models but also, significantly, by modern neural networks with ReLU activations.

**Questions:**

Please see the above sections.

Q1: How robust is the “piecewise-constant” nature of the overall cost function $V(I,w)$ to the choice of the penalty functions $P_k$? Specifically, if $P_k$ was not constant but a complex function modeling the time of an action (e.g., the time to solve an LP), would the core theoretical result (Theorem 3.3) still hold?

Q2 : In Proposition 3.8 you deliberately use the weakened upper bound on $\Gamma$ from Lemma 3.7, even though the proof of Lemma 3.7 yields a strictly tighter bound. Why not employ the tighter bound to obtain a sharper estimate of the Pdim in Proposition 3.8?

**Ethical Concerns:**

["NO or VERY MINOR ethics concerns only"]

**Final Justification:**

I recommend accepting this paper. The authors have addressed most of my concerns: they clarified how their bounds act as practical “scaling laws,” added a clear data-dependent Rademacher analysis, showed how to extend their work beyond ReLU activations, and provided experiments that support their theory. These improvements strengthen both the theoretical and practical impact of the paper and outweigh the few remaining minor issues.

**Limitations:**

yes

**Quality:**

3

**Strengths And Weaknesses:**

Though I checked the proofs, I am not an expert in statistical learning theory and may have overlooked some concepts

# Strengths
1. The primary strength of this paper is extending the formal analysis of learnable algorithm policies to the non-linear domain, specifically ReLU-based MLPs. This is highly relevant, as much of the recent empirical success in this area has come from such models. This work provides a much-needed theoretical foundation for these practices.

2. The introduction of the $(\Gamma,\gamma,\beta)$-structure is an elegant and powerful abstraction.

3. The paper is mathematically rigorous. All major claims are stated formally as theorems, propositions, or lemmas, and are accompanied by detailed proofs in the appendix.

# Weaknesses:

1. The primary weakness is that the derived worst-case bounds (Propositions 3.8, 4.1, 4.2) depend on parameters that can be astronomically large, such as the maximum number of algorithm rounds \(M\) and the total number of network parameters \(W\). The dependency on \(M\) is particularly concerning, as it can lead to exponentially large terms in the complexity bounds. This raises questions about the practical utility of the bounds, as they may be too loose to provide meaningful sample size estimates for real-world applications.

2. The analysis for neural networks is contingent on the activation functions being piecewise-polynomial. This is a strong restriction that excludes many popular and high-performing non-linearities (e.g.,  Sigmoid) and advanced architectures (e.g., GNNs, attention mechanisms). This limits the direct applicability of the results to a specific, albeit common, class of neural network architectures.

3. No experiments are presented to gauge how the theoretical bounds correlate with actual B&C policy learning e.g., how many instances are needed in practice, or whether data-dependent Rademacher estimates are indeed small. This leaves a gap between theory and solver practice.

## Minor
4. The title, "Generalization Guarantees for Learning Branch-and-Cut Policies," implies broad applicability. However, the concrete results for neural networks apply to a very specific class of models. This creates a potential mismatch between the perceived scope of the paper and the actual scope of the theoretical guarantees provided. While all theoretical work relies on assumptions, the restrictiveness here is significant enough that the title may be seen as over-promising.

---

> ### Author Rebuttal · Authors · 2025-07-30
>
> Thank you very much for reading our paper and providing detailed feedback.
>
> **Weaknesses:**
>
> 1. We thank the reviewer for this insightful comment and for raising an important point about the nature of worst-case theoretical bounds. We would like to begin by clarifying the primary message of our bounds for practitioners. The main takeaway is not to estimate a precise sample size from the worst-case formula, but rather to provide a "scaling law" for neural network design in this setting. The key insight from our analysis is that the sample complexity scales linearly with the number of network parameters ($W$) and layers ($L$), as shown in Propositions 3.8, 3.9, 4.1 and 4.2. This provides direct, actionable guidance: if a practitioner has a trained model that already achieves a satisfactory performance, but wishes to increase its expressive power with a larger architecture to further reduce the inductive bias, our results indicate that maintaining a similar generalization guarantee requires that the number of training instances increase by a commensurate factor. This theoretical principle, derived from our specific analysis of the sequential decision process, establishes a clear and practical trade off between model complexity and data requirements in this domain.
>
>     Having said that, we agree that the pseudo-dimension bounds, by design, provide a distribution-independent guarantee for any problem fitting our general framework, and can thus appear loose due to the dependency on worst-case parameters like $M$. We address this in the paper in two ways and Appendix C.2 is dedicated to this.
>     * Our work also provides a path to tighter, data-dependent guarantees via the empirical Rademacher complexity (Proposition 3.9), and we refer the reviewer to Appendix C.2 for a detailed discussion illustrating the advantages of this approach. To make this concrete, this discussion includes an example analyzing a class of integer programs with $n$ variables where branching actions can lead to rapid termination of the search tree. In this setting, the number of states encountered for an instance is merely $O(n)$. The pseudo-dimension bound, however, must account for the worst case and thus scales with a term like $\sqrt{\frac{M\log n}{N}}$, where $N$ is the training sample size and $M$ could be as large as $\Omega(2^n)$. In contrast, our Rademacher bound adapts to the problem's benign structure and scales with a much smaller term, $\sqrt{\frac{\log(Nn)}{N}}$, offering a much tighter and more meaningful guarantee.
>     * Even for the worst case pseudo-dimension based bounds, we discuss in Appendix C.2 how the the structure of the specific decision making process could imply a pseudo-dimension bound that is much better than something that scales with $M$. In particular, the factor $M$ really comes from having to count all possible paths that the decision process could take, which could be exponential in $M$. In contrast, the pseudo-dimension can alternatively be bounded by other parameters; for example, the total number of states which could be an exponential factor smaller than the total number of paths (something that is exploited by dynamic programming for solving such decision making problems). See the discussion in lines 692-700 in Appendix C.2.
>
>     We did not include this discussion in the main text so as to not distract from the main message, which is to show that bounds for general decision making processes can be obtained with immediate consequences for B&C. We thought the fine-tuned bounds were better left in an appendix for an interested reader. We aim to produce a journal version of this paper that will be more detailed and nuanced in its discussions.
>
> 2. This is an excellent point, and we thank the reviewer for highlighting the scope of our current analysis. We chose to focus on piecewise polynomial scoring functions in this work for several reasons: they form a natural and non-trivial generalization of the linear models studied in prior work, and this class is already rich enough to provide the first generalization guarantees for widely-used architectures like Multi-Layer Perceptrons (MLPs) and Graph Neural Networks (GNNs) (with a fixed number of message passing rounds) with piecewise polynomial activations.
>
>     However, the reviewer is correct that this excludes other important non-linearities. Nevertheless, our proof technique is more general and can be extended to accommodate neural networks with exponential-based activations (e.g., Sigmoid, Tanh, Swish) and, crucially, advanced architectures involving attention and Softmax mechanisms.
>
>     This extension requires introducing the theory of Pfaffian functions [1, 2], as modern architectures—such as the composite GNN, attention, and MLP model in [3]—result in scoring functions that exhibit a "piecewise Pfaffian" structure. A recent, concurrent work [4] has also laid groundwork for PAC learning with this specific structure. The analytical path for this extension involves replacing the sign pattern counting argument from our Lemma A.2. Doing this for Pfaffian structures is founded on a powerful result by Khovanskiĭ [1], analogous to Bézout's theorem for polynomials, that provides an explicit bound on the number of non-degenerate solutions to a system of Pfaffian equations. This allows us to bound the number of connected components for any intersection of the decision boundaries defined by Pfaffian structures. Following Lemma 7.9 of Anthony \& Bartlett [5], this bound on intersections can be used to derive an upper bound on the number of sign patterns generated by the decision boundaries.
>
>     Following this path, the central structural result of our paper would still hold: if the scoring functions are piecewise Pfaffian, the final cost function $V(I,\mathbf{w})$ is still a piecewise constant function of $\mathbf{w}$ for any fixed $I$, with boundaries defined by Pfaffian structures, and corresponding pseudo-dimension and Rademacher complexity bounds can be derived. This would handle nonlinear activations like Sigmoid, as well as models like GNNs composed with attention mechanisms. We will add a dedicated section in the final version of the paper outlining this extension and its implications.
>
> 3. We did not include experiments in the original submission as our primary motivation was to provide the theoretical foundations for the successful empirical works surveyed in Section 2.1. However, we agree with the reviewers that a computational study strengthens the paper. We have thus conducted a new proof-of-concept experiment to provide an empirical view of our theoretical bounds. The results verify the predicted $\mathcal{O}(1/\sqrt{N})$ convergence rate ($R^2=0.944$) and show that practical generalization errors are substantially smaller than the worst-case bounds. For the full details of this experiment, we respectfully refer the reviewer to our response to Reviewer U6oB.
>
> **Title.** Indeed, the title refers to the broad applicability of our theoretical framework, which extends beyond the specific models we instantiate. While our concrete results focus on the class of MLPs with piecewise polynomial activations, the framework itself is more general. As detailed in our response to Weakness 2, our analysis can be extended to the broader class of piecewise Pfaffian functions, providing guarantees for modern architectures with Sigmoid and attention. To improve clarity, we will revise the abstract to specify that our main results are instantiated for piecewise polynomial models, and keep the title to reflect the framework's full potential.
>
> **Questions:**
> 1. Yes, our core theoretical result (Theorem 3.3, which states no requirements for $P_k$) holds for any choice of penalty functions $P_k$, including complex, non-constant ones like local computation time. The reason is that our analysis partitions the parameter space $\mathcal{W}$ into regions within which the entire execution trace—the full sequence of states visited and actions taken—is invariant. This partition is determined exclusively by the scoring functions. Since the execution trace is fixed within each region, the accumulated cost, $V(I,\mathbf{w})=\sum P_k(s,a^*,i)$, is a sum over a fixed sequence of penalty values and is therefore constant, regardless of the functional form of $P_k$. The only assumption needed is that an identical action in an identical state incurs an identical cost.
> 2. This is a sharp observation, and we thank the reviewer for their careful reading. Our decision to use the simplified bound was a deliberate choice to prioritize the clarity and interpretability of the final pseudo-dimension bound in Proposition 3.8. The simplified form allows a reader to immediately see the asymptotic dependencies on high-level network parameters ($L,W,U$), rather than being obscured by more complex summation terms. While this simplification loses some precision in the constants, it does not alter the core conclusion of our analysis. Per the reviewer's valuable suggestion, we will add a remark in the final version to acknowledge the tighter intermediate bound and explicitly discuss this trade-off between algebraic simplicity and tightness.
> ---
> [1] Khovanskiĭ, A.G., 1991. Fewnomials (Vol. 88). American Mathematical Soc..
>
> [2] Karpinski, M. and Macintyre, A., 1997. Polynomial bounds for VC dimension of sigmoidal and general Pfaffian neural networks. Journal of Computer and System Sciences, 54(1), pp.169-176.
>
> [3] Paulus, M.B., Zarpellon, G., Krause, A., Charlin, L. and Maddison, C., 2022, June. Learning to cut by looking ahead: Cutting plane selection via imitation learning. In International conference on machine learning (pp. 17584-17600). PMLR.
>
> [4] Balcan, M.F., Nguyen, A.T. and Sharma, D., Algorithm Configuration for Structured Pfaffian Settings. Transactions on Machine Learning Research.
>
> [5] Anthony, M. and Bartlett, P.L., 2009. Neural network learning: Theoretical foundations. Cambridge University Press.

---

> > ### Comment · Reviewer_SAVB · 2025-08-06
> >
> > Thank you for your detailed rebuttal. You’ve addressed my concerns about the worst-case bounds, data-dependent Rademacher estimates, and extension to general activations. I believe this is a strong paper and have raised my score accordingly.

---

> > > ### Author Response · Authors · 2025-08-06
> > >
> > > Thank you for your careful reading and thoughtful feedback. We’re glad our clarifications helped and we appreciate your updated assessment.

---

### Note · Authors · 2025-08-13

We express our sincere gratitude to the reviewers for their thoughtful engagement and constructive feedback. We are very encouraged by the positive reception of our work. Reviewers characterized our theoretical framework as "mathematically rigorous" and "elegant" (SAVB), and a "strong theoretical contribution" (U6oB). We particularly appreciate the recognition that our work "contributes a missing piece" (twK1) by extending generalization analysis to the non-linear domain (SAVB), thereby "bridging the gap between empirical methods and theoretical guarantees" (u1pW).

We believe we have successfully addressed the reviewers' comments during the discussion phase. We summarize the main clarifications and additions below:

First, responding to the suggestions on empirical validation (SAVB, U6oB, twK1), we conducted a new proof-of-concept experiment analyzing a learned cut-selection policy. The results verify the convergence rate suggested by theories, and confirm that practical generalization errors are much smaller than the worst-case bounds. This study will be added to the final version.

Second, we clarified the interpretation and tightness of our bounds (SAVB, u1pW). We emphasized that the primary utility of the worst-case bounds is the "scaling law" they establish: sample complexity scales linearly with network parameters ($W$) and layers ($L$). This provides a guidance on the trade-off between model complexity and data requirements. Furthermore, we highlighted our data-dependent bounds (Prop 3.9 and Appendix C.2), which can offer significantly tighter guarantees by adapting to the actual complexity of the instances, avoiding worst-case dependencies.

Third, regarding the scope of the analysis (SAVB), we detailed how our techniques extend beyond piecewise-polynomial activations to the broader class of "piecewise Pfaffian" functions. This extension covers modern architectures involving exponential-based activations (e.g., Sigmoid) and, by extension, the models that rely on them (e.g., attention mechanisms). We will add a dedicated section outlining this extension, including specific case studies as suggested by u1pW.

We believe these additions and clarifications have significantly strengthened the paper, better connecting our theories with solver practice. We are pleased that the reviewers recognize this as a "nice theory paper" (twK1), a "strong paper" (SAVB), and a "high-impact paper" (U6oB), and we thank them again for their invaluable input.

---

### Decision · Program_Chairs · 2025-09-17

**Decision:**

Accept (poster)

**Comment:**

This paper studies the learning of score-based policies for branch-and-cut methods in MIP solving. The authors show a uniform convergence result for score functions across different models (say, the space of NN parameters for a fixed architecture), by bounding standard learning-theoretic quantities (e.g. pseudo-dimensions for data-independent bounds, and Rademacher complexity for data-dependent bounds).

Reviewers and I generally liked the result: it provides a solid theoretical foundation for a relevant and important component in MIP solving, and is a step towards bridging theory and empirical methods.

Here are two concrete suggestions from the reviewer discussion:
- Please consider adding the phrase "score-based" somewhere in the title. The current title is overclaiming a bit.
- Please include the experimental results requested by Reviewer U6oB, ideally in the main body using the extra page.